

# Traces of urban forest in temperature and $CO_2$ signals in monsoon East Asia

Keunmin Lee[1], Je-Woo Hong[2], Jeongwon Kim[1], and Jinkyu Hong[1]

[1]Department of Atmospheric Sciences, Yonsei University, Seoul, 03722, Korea (Republic of)

[2]Korea Environment Institute, Sejong, 30147, Korea (Republic of)

*Correspondence to*: Jinkyu Hong (jhong@yonsei.ac.kr)

**Abstract.** Cities represent a key space for our sustainable trajectory in a changing environment, and our society is steadily embracing urban green space for its role in mitigating heatwaves and anthropogenic $CO_2$ emissions. This study reports two-year surface fluxes of energy and $CO_2$ measured via the eddy covariance method in an artificially constructed urban forest to examine the impact of urban forests on air temperature and net $CO_2$ exchange. The urban forest site shows typical seasonal patterns of forest canopies with the seasonal march of the East Asian summer monsoon. Our analysis indicates that the urban forest reduces both the warming trend and urban heat island intensity compared to the adjacent high-rise urban areas and that photosynthetic carbon uptake is large despite relatively small tree density and leaf area index. During the significant drought period in the second year, gross primary production and evapotranspiration decreased, but their reduction was not as significant as those in natural forest canopies. We speculate that forest management practices, such as artificial irrigation and fertilization, enhance vegetation activity. We also stipulate that ecosystem respiration in urban forests is more pronounced than typical natural forests in a similar climate zone. This can be attributed to the substantial amount of soil organic carbon available due to intensive historical soil use and soil transplantation during forest construction, as well as relatively warmer temperatures in urban heat domes. Our observational study also indicates the need for caution in soil management for less $CO_2$ emissions in urban areas.

## 1 Introduction

Cities inhabit only 2% of the Earth's land surface but hold more than 55% of the world's population. With the unprecedented rapid urbanization in the last century, our life trajectory heavily depends on urban structures and functions, and it is expected that the urban population will increase by up to 68% by 2050 (UN, 2019). Our current concern is regarding the disastrous impacts of climatic events (e.g., heatwaves, flooding, and drought) and environmental changes (e.g., air pollution and land degradation) on our socioeconomic system in a changing climate (McCarthy et al., 2010; Rahmstorf and Coumou, 2011). Accordingly, it remains an urgent issue to implement integrated policies for climate change mitigation and adaption toward sustainable cities against global warming and related natural disasters. In particular, urban green infrastructures, such as urban forest, have been recognized as a key solution toward alleviating climatic and environmental disasters (e.g., Oke et al., 2017; Chiesura, 2004; Haaland and van den Bosch, 2015; Kroeger et al., 2019). Green spaces in cities, as opposed to gray spaces, are exposed to wide ranges of environmental and climatic conditions across geographical locations.



Especially when green spaces replace gray infrastructures during the urban redevelopment, it remains unclear
whether their benefits emerge in real conditions and thereby outperforming their maintenance cost and other
harmful effects. To leverage their full potential benefits, it is necessary to assess the biophysical effects of urban
forests based on direct long-term monitoring in urban areas.
Urban forests are a key part of green infrastructures in a city, and two of their benefits, which have been mainly
addressed in previous studies, are thermal mitigation and carbon uptake (Roy et al., 2012; Oke et al., 2017). Firstly,
urban forests mitigate direct sunlight and diminish the incoming radiant energy on the land surface, thereby
reducing surface temperature. Additionally, urban forests supply water to the atmosphere through transpiration
and retain water for longer times than the impervious surfaces of urban structures. These processes contribute
toward reducing air temperature by partitioning more available energy to latent heat flux ($Q_E$) than sensible heat
flux ($Q_H$), thus creating favorable conditions for mitigating heatwaves and related health problems (e.g., Oke,
1982; Hong et al., 2019a). Eventually, this cooling effect reduces the electrical energy load of buildings, as well
as greenhouse gas emissions. Previous studies have reported cooling effects of urban forests from street trees to
parks scales (Oke et al., 1989; Bowler et al., 2010; Norton et al., 2015; Shashua-Bar and Hoffman, 2000). Such
cooling effects depend not only on tree species and structures (Feyisa et al., 2014) but also on the size and
vegetation density of urban green areas (Yu and Hien, 2006; Chang et al., 2007; Hamada and Ohta, 2010; Feyisa
et al., 2014). However, despite the strong temperature-controlling factors of evapotranspiration (ET) and direct
heat fluxes over urban forest canopies, only a few studies have reported surface energy balance (SEB) in urban
forests in relation to thermal mitigation based on direct measurements (e.g., Oke et al., 1989; Spronken-Smith et
al., 2000; Coutts et al., 2007a; Ballinas and Barradas, 2015; Hong and Hong, 2016;). Moreover, it is noticeable
that forest cooling intensity depends on geography and even forests can produce a warming trend with the
decreased albedo (Bonan, 2008; Wang et al., 2018). The lack of direct urban forest measurements hinders proper
assessment of their influences on the climate and environment.
Furthermore, urban forests mitigate anthropogenic carbon emissions by photosynthetic $CO_2$ uptake. Traditionally,
carbon uptake by urban forests has been estimated by empirical relationships (e.g., biomass allometric equation)
or short-term inventory of biomass data and vegetation growth rates, which have limitations of spatiotemporal
coverage (Rowntree and Nowak, 1991; Nowak, 1993; Nowak et al., 2008; Weissert et al., 2014). Currently, the
eddy covariance (EC) method is being applied in various ecosystems from grasslands and natural forests to urban
areas because it provides continuous net $CO_2$ flux measurements at the neighborhood scale every half hour
(Christen 2014). From this perspective, the EC method is useful for studying the net $CO_2$ exchange ($F_C$) from
diurnal to interannual variations, with its simultaneous measurement of surface energy fluxes. Recently, direct $F_C$
measurements have been performed using the EC method in urban green spaces to examine SEB and carbon
exchange (Coutts et al., 2007a, 2007b; Awal et al., 2010; Bergeron and Strachan, 2011; Crawford et al., 2011;
Kordowski and Kuttler, 2010; Peters and McFadden, 2012; Velasco et al., 2013; Ward et al., 2013; Ueyama and
Ando, 2016; Hong et al., 2019b; Hong et al., 2020). However, the EC method provides only the net effects of $CO_2$
exchange from various carbon sources and sinks, which limits the physical interpretation and assessment of the
benefits and costs of urban forests.



Flux partitioning into photosynthesis and ecosystem respiration from the EC measured $F_C$ requires additional
information and data processing (Stoy et al., 2006). It is more challenging to partition $F_C$ into individual sources
and sinks, particularly in urban areas because of the complex contributions from biogenic (e.g., vegetation
photosynthesis, respiration of vegetation, soil, and humans) and extra anthropogenic (e.g., fossil fuel combustion
by transportation or in households and commercial buildings) processes (Pataki et al., 2003). Stochastic $F_C$
partitioning methods were recently applied by reprocessing EC observation data with auxiliary data and provided
useful knowledge on urban carbon cycle (Hiller et al., 2011; Crawford and Christen, 2015; Menzer and McFadden,
2017; Stagakis et al., 2019).
With this background, the objectives of this study include: 1) reporting temporal changes in air temperature after
the artificial construction of an urban forest park in the Seoul metropolitan area where a hot and humid summer
season affects and shows steep global warming trends (Hong and Hong, 2016) and 2) quantifying the carbon
uptake of urban forests based on the $F_C$ partitioning through the data observed by the EC method and
meteorological data (Lee et al., 2021). Here, we highlight the biotic and abiotic factors controlling the carbon
cycle in urban forests and the impact of urban forests on the thermal environment after forest park construction.
**2 Materials and Methods**
**2.1 Urban surface energy and $CO_2$ balances**
The SEB is expressed as:

89         $Q^* + Q_F = Q_H + Q_E + \Delta Q_S + \Delta Q_A$      (1)

where $Q^*$ is the net all-wave radiation of the sum of outgoing and incoming short- and long-wave radiative fluxes,
$Q_F$ is the anthropogenic heat flux, $Q_H$ is the turbulent sensible heat flux, $Q_E$ is the latent heat flux, $\Delta Q_S$ is the net
storage heat flux, and $\Delta Q_A$ is the net heat advection (Definitions of variables in Appendix A).
The surface $CO_2$ budget in an urban forest is formulated as follows:

94         $F_C = E_R + E_B + RE - GPP \equiv E_R + E_B + NBE$      (2)

where $F_C$ is the net $CO_2$ exchange at the city-atmosphere interface, $E_R$ and $E_B$ are the anthropogenic $CO_2$ emissions
from fossil fuel combustion by vehicles and heating in a building, respectively. *GPP* and *RE* are biotic
contributions to $F_C$; *GPP* is the gross primary production as a result of photosynthetic $CO_2$ uptake, and *RE* is the
ecosystem respiration from soil and vegetation. Urban ecosystem respiration considers not only the autotrophic
and heterotrophic respirations of vegetation and soil but also human respiration (Moriwaki and Kanda, 2004;
Velasco and Roth, 2010; Ward et al., 2013, 2015; Hong et al., 2020). Human respiration is negligible in this study
because there is no residential population in the park. Vegetation in urban areas includes trees and lawns in urban
forests, as well as gardens and roadsides, and it offsets $CO_2$ emissions through $CO_2$ assimilation by photosynthesis
as the only carbon sink.





Additionally, *NBE* is the net biome $CO_2$ exchange and is typically defined as the net ecosystem exchange by *RE*
– *GPP* for natural vegetation. Put differently, *NBE* refers to carbon losses in heterotrophic respiration minus the
net primary production on natural vegetative surfaces; thus, negative *NBE* indicates the net carbon uptake by the
natural ecosystem (Kirschbaum et al., 2001; Randerson et al., 2002). Unlike natural ecosystems, the $F_C$ between
an urban forest and atmosphere is a complex mixture of biogenic (i.e., *GPP* and *RE*) and anthropogenic (i.e., $E_R$
and $E_B$) processes across various spatial and temporal scales. In urban environments, anthropogenic emissions
depend on the local characteristics (e.g., climate, population density, levels of industrial activity, and existing
carbon intensity of electricity supply) of the city and locations of the eddy covariance system (Feigenwinter et al.,
2012; Kennedy et al., 2014; Lietzke et al., 2015; Stagakis et al., 2019).

**2.2 Site description**
**2.2.1 Seoul Forest Park**
Micrometeorological measurements were taken at the Seoul Forest Park (SFP) in the Seoul metropolitan area,
Korea (37.5446°N, 127.0379°E). SFP is the third largest park in Seoul with an area of 1.16 km$^2$ (Fig. 1a). This
area had been used as a horse racetrack and a golf course inside the track since 1950 and was surrounded by
cement factories to the west (Fig. 1b). The local government initially planned this area as a commercial district
with a high-rise multi-purpose building complex but changed its plan to redevelop the area as a green space in
late 1990s. The construction of the SFP began in December 2003, and it was opened to the public in June 2005
(Fig. 1c).
The dominant land cover within a 300-m radius of the measurement system is a deciduous forest with irrigated
grass lawns (*Zoysia*), oak (*Quercus acutissima*), ginkgo (*Ginkgo biloba*), and ash trees (*Fraxinus rhynchophylla*),
which correspond to the Local Climate Zone (LCZ) 'A', dense trees (Stewart and Oke, 2012). The maximum leaf
area index (LAI) of $300 \times 300$ m$^2$ around the SFP tower is approximately 1.6 (Copernicus Service information,
2020). On the east side (0–120°), there are trees (approximately 230 stems ha$^{-1}$) with a small artificial lake and
grasslands beyond it. Trees mainly occupy the southern and western directions of a tower (120–330°) within a
100-m radius area (approximately 540 stems ha$^{-1}$) and traffic roads lie outside of the dense vegetation. The mean
tree height ($h_c$) is approximately 7.5 m and ranges between 5.8–9.5 m. The mean roughness length ($z_0$) and zero-
plane displacement height ($z_d$) are estimated by the tree height-based approach within 100 m radius and they range
between 0.3–0.6 and 4.1–8.2 m, respectively (Raupach et al., 1991). $z_0$ and $z_d$ have seasonal and directional
variations depending on the variability of the leaves on the vegetation (Lee, 2015; Kent et al., 2018). $z_0$ and $z_d$
change from approximately 0.6 and 5.0 m during leaf-on period (June–August) to 1.2 and 3.0 m during the leaf-
off periods (December–February) by the Macdonald method (Macdonald et al., 1998). Approximately 80% of the
footprint area of the SFP tower is within 250 m (Fig. 1e).
The traffic roads consist of eight and ten lanes carrying heavy traffic throughout the day (~100,000 vehicles day$^{-1}$
$^{1}$) to the south and west of the tower (Fig. 1c). Hourly traffic volume, which is used for surface flux partitioning,
is evaluated on the road adjacent to the SFP tower every year by the Seoul Metropolitan Government



(https://topis.seoul.go.kr). Across the road on the western side of the tower, a cement factory still exists, although
its size is smaller than it used to be in the past (Fig. 1b and 1c).

**2.2.2 Climate conditions**

Climatic condition shows a distinct seasonal variation with the seasonal march of the East Asian summer monsoon.
The mean climatological values (1981-2010) of the screen-level air temperature ($T_{air}$) and precipitation were
12.5°C and 1450 mm year$^{-1}$, respectively. During the study period (June 2013–May 2015), the observed $T_{air}$ was
higher than the climatological mean. Higher temperatures lasted longer in the summer of 2013 with the stagnation
of the migratory anticyclones (June) and North Pacific anticyclone (July–August). There were strong heatwaves
in the spring seasons of 2014 and 2015 (Hong et al., 2019a).
Notably, seasonal precipitation shows a contrasting pattern between two consecutive years (Fig. 2d). In the first
year (June 2013–May 2014), annual precipitation was 1256 mm, which corresponded to approximately 90% of
the climatological mean. In addition, approximately 50% of the annual rainfall was concentrated in the summer
with an estimated 650 mm occurring only in July 2013; however, in the second year the annual rainfall was 932
mm (i.e., 67% of the climatological mean). The monthly precipitation values in the July and August of 2014 were
198 and 169 mm, respectively, which represented only approximately 35% of the climate mean. Accordingly, the
vapor pressure deficit ($VPD$) and downward shortwave radiation ($K_{\downarrow}$) in July 2013 were relatively smaller than
those in July 2014 (Fig. 2b and 2c).

**2.2.3 Observations in the Seoul Metropolitan Area**

In this study, meteorological data from six stations (one eddy covariance station, one aerodrome meteorological
observation station, and four automatic weather stations) in the Seoul Metropolitan Area are additionally analyzed
to examine the heat mitigation and $CO_2$ reduction effects of urban vegetation in the SFP (Table 1 and Fig. 1a).
The Eunpyeong eddy covariance site (EP, 37.6350°N, 126.9287°E) is for surface flux observations in the
northwest of Seoul, where there was a recent urban redevelopment to high-rise and high-population residential
areas from low-rise areas (Hong and Hong, 2016; Hong et al., 2019b). Flux observations at the site have been
conducted since 2012, and they show the SEB and turbulence characteristics of a typical urban residential area.
Because the area around the SFP was originally planned to be redeveloped to high-rise high-population residential
buildings, EP is selected for comparative analysis as a hypothetical place for the SFP region because they are
close to each other and so have the similar synoptic conditions.
The Gimpo Airport Observatory (GP, 37.5722°N, 126.7751°E) is located on the western boundary of Seoul, and
it is surrounded by grasslands and croplands, which corresponds to LCZ 'D'. As the dominant wind comes from
the west, the GP site is generally affected by the same synoptic weather conditions as Seoul. The GP station
represents the rural environment of the Seoul Metropolitan Area because urban development is restricted around
the airport (Hong et al., 2019a). In this study, we select the GP site as a reference point and calculate the urban
heat island intensity (UHIi) as the synchronous difference in $T_{air}$ between the urban and rural areas accordingly
(Stewart, 2011).



The Seongdong Observatory (SD, 37.5472°N, 127.0389°E), the closest station to the SFP, is located
approximately 300 m north of the SFP tower (Fig. 1c). Since the station began observations in August 2000, the
meteorological data at SD are useful for analyzing temperature changes before and after the construction of the
SFP. Accordingly, it is used to analyze local climatic changes caused by the SFP. Moreover, SD provides auxiliary
weather variables (e.g., precipitation) that are not observed in SFP station and reference data for surface flux gap
filling. The Gangnam, Seocho, and Songpa observatories (hereafter denoted as AVG) are located in Seoul's
central business district, which corresponds to LCZ 1 or 2. These sites are also close to the SFP (~ 5 km); thus,
temperatures in these regions can be assumed to be exposed to the same synoptic condition. These regions show
greater UHIi than other parts of Seoul because of dense skyscrapers, according to the analysis of the spatial
distribution of UHIi in Seoul (Hong et al., 2013). The average temperature of these three automatic weather
stations is used to evaluate the temperature and UHIi reduction effects of the SFP construction. All meteorological
data from the automatic weather station and aerodrome meteorological observation station are observed every
minute, and they are averaged for 1 h for UHIi analysis. All the meteorological data are processed for quality
control on the National Climate Data Portal of the Korea Meteorological Administration (http://data.kma.go.kr).
**2.3 Instrumentation and data processing**
The measurement system was installed on the rooftop of the SFP facility building (Fig. 1d). A three-dimensional
sonic anemometer (CSAT3A, Campbell Scientific, USA) and enclosed infrared gas analyzer (EC155, Campbell
Scientific, USA) were mounted 12.2 m above the ground level (2.8 m above the roof of an 8.4 m high building)
for 2 years (Fig. 1d). The eddy covariance data were recorded using the data logger (CR3000, Campbell Scientific,
USA) with a 10-Hz sampling rate and a 30-min averaging time. The gas analyzer was calibrated with standard
$CO_2$ gas every three months. The main footprint covered the forest canopies, and the measurement height ($z_m$)
satisfied the tower height requirement over forested or more structurally complex ecosystems (i.e., $z_m \cong z_d + 4(h_c$
$- z_d)$) (Munger et al., 2012). Two radiometers (NR Lite2 and CMP3, Kipp&Zonen, Netherlands) were used to
measure the radiative fluxes. An auxiliary measurement included a humidity and temperature probe (HMP155A,
Vaisala, Finland).
The 30-min flux is computed using EddyPro (6.2.0 version, LI-COR), with the applications of the double rotation,
time lag compensation using covariance maximization, spike removal and quality test (Vickers and Mahrt, 1997),
spectral corrections for low-frequency (Moncrieff et al., 2004) and high-frequency (Fratini et al., 2012), as well
as vertical sensor separation correction (Horst and Lenschow, 2009). We apply the following post processes for
quality control: 1) plausible value check, 2) spike removal, and 3) discarding the negative $F_C$ flux during the
nighttime (Hong et al., 2020). The total study period from installation (31 May, 2013) to termination (03 June,
2015) is approximately 2 years (35,174 potential 30-min data), and the total available data are approximately
90.1%, 88.3%, and 85.4% (n = 31709, 31064, 30028) for $Q_H$, $Q_E$, and $F_C$ after the processes, respectively.
It is important to partition the $F_C$ into four contributing components (i.e., $RE$, $GPP$, $E_R$, and $E_B$ in Eq. 2) to
investigate their biotic and abiotic controlling factors in an artificially constructed park. This study applies for a





statistical partitioning method described in Lee et al. (2021). More information and relevant figures on the flux
partitioning are available in Lee et al. (2021).

**3 Results and discussion**

**3.1 Surface energy balance**

The SEB at the SFP shows typical seasonal variations over natural forest canopies with the seasonal march of the
East Asian monsoon (Fig. 4) (Hong and Kim, 2011; Hong et al., 2019b; Hong et al., 2020). In summer, there are
lengthy rainy spells and large temporal variabilities of meteorological conditions with the impacts of the East
Asian summer monsoon (Fig. 2d). This heavy rainfall causes substantial decreases in $K_\downarrow$, and thus $Q^*$, with large
temporal variations, thereby leading to the mid-summer depression of surface fluxes (Fig. 2c and 4). $Q^*$ also
reaches its maximum in spring rather than in summer and decreases gradually from spring to winter (Fig. 4). More
than half of $Q^*$ is partitioned to $Q_E$, and $Q_H$ is minimum in summer owing to the ample water supply from the
summer rainfall. However, $Q_H$ is maximum in spring and even larger in winter, despite the relatively smaller $Q^*$,
because of the cold and dry climatic conditions induced by the winter monsoon. Accordingly, the seasonal mean
Bowen ratio ($\beta = \sum Q_H / \sum Q_E$) ranges from near zero (summer) to approximately 4 (winter) with its daily maximum
around 9 in early January 2015 (Fig. 5)**.** Notably, $\beta$ in the SFP is consistently lower than the high-rise, high-density
residential area (i.e., the EP site) because of the ET from the vegetative canopies and the unpaved surfaces in the
urban forest. This difference between the two distinct sites confirms that urban forests are responsible for
substantial changes in the thermal environment in terms of $Q_H$ and $Q_E$, as well as their related air and surface
temperatures because of more evaporative cooling in green spaces compared to impervious surfaces such as roads
and buildings in urban areas (Oke et al., 2017).
The SEB also shows interannual variabilities over forest canopies influenced by the timing of the onset and
duration of the summer monsoon (Hong and Kim, 2011) (Fig. 6). As discussed in Section 2.1.2, annual
precipitation is much larger in the first year than in the second year because of the interannual variations in the
East Asian monsoon activity, thereby making substantial differences in surface radiative fluxes. Furthermore, $Q_E$
shows the difference between the first and second years of the observation, particularly by responding to such
interannual variability of radiation. In the first year of the observation, $Q_E$ is more than 300 W m$^{-2}$ and has a
relatively larger temporal variability because of the frequent rainfall events in summer, compared to the second
year.
Evapotranspiration rate ranges from 5 mm month$^{-1}$ in January 2015 to 74 mm month$^{-1}$ in August 2013, and the
annual ET values are 367 and 320 mm year$^{-1}$ in the first and second years, respectively (Fig. 5 and Table. 2). The
ET values correspond to 29.3% and 34.3% of the annual precipitations (Fig. 2d). The difference in ET between
the two consecutive years (i.e., 48 mm) mainly occurred in summer (42 mm), especially in August (30 mm). It
has been reported that approximately 55% of the net radiation is partitioned to latent heat flux in forest canopies
globally (Falge et al., 2001; Suyker and Verma, 2008). The annual ET to net radiation from the urban forest is
smaller than this global average and it is also smaller than that of forests at similar latitudes in the East Asia



(Khatun et al., 2011). The annual ET in the second year is smaller than that in the first year with extensive drought
in the second year. However, it is notable that the ET in the second year shows only an approximately 12%
decrease compared to the first year, despite a substantial decrease in precipitation (26% decrease) and similar net
radiation (Table 2). Although the summer monsoon provides ample water to the ecosystem, its delay and weakness
result in severe drought and stress to the ecosystem in this region (Hong and Kim, 2011); however, such ecosystem
stress, such as the shrinking of ET and carbon uptake, is inexplicit for the urban forest. We speculate that artificial
irrigation by a sprinkler mitigated ecosystem stress to a certain degree in the urban forest.

### 3.2 Air temperature

Figure 6 shows the mean diurnal pattern of the air temperature difference between the AVG and SD near the SFP
($\Delta T_{air} \equiv T_{air\_AVG} - T_{air\_SD}$ hereafter) before and after the park construction in summer. Notably, $\Delta T_{air}$ is always
positive during the entire summer season (i.e., AVG is warmer than SD) and shows distinct impacts in terms of
magnitude and diurnal variations after the park construction. The warming trend is evident at the AVG (p<0.015),
wherein there were no changes in the urban structure and function around them. The warming rate at the AVG is
3.0 °C century$^{-1}$, which corresponds to the warming rate reported in the high-rise urban area in Seoul (Hong et al.,
2019a). However, the warming rate around the SFP is approximately 1.6 °C century$^{-1}$, which is smaller than that
of the AVG and other urban areas in Seoul and is comparable to the global mean warming rate of 0.9 °C century$^{-}$
$^{1}$ (Hansen et al., 2010; Hong et al., 2019a).
Notably, such a lower warming trend around the SFP mainly occurs in the afternoon when ET is dominant. This
difference will be larger if we consider that the measurement height at the AVG is higher than that at the SD
(Table. 1). The maximum $\Delta T_{air}$ is approximately 0.3 °C around 10:00 before the park construction (Fig. 6a) and
increases up to approximately 0.5 °C with its peak occurrence shifting from the morning to the afternoon (i.e.,
around 14:00) after the construction (Fig. 6b). This peak time in the afternoon is coincident with the time when
photosynthesis is the highest in the vegetation; thus, $Q_E$ increases in summer. Our results indicate that the thermal
mitigation of the urban forest is important as a result of increases in ET, especially if we consider that the SFP
area was originally planned to be developed as a high-population multipurpose building complex.

### 3.3 Urban heat island intensity

The influence of urban forests on summer temperature produces also evident traces in UHIi. Figure 7 shows the
mean diurnal variation of UHIi at the SFP and AVG during summer. Apparently, the UHIi of the SFP (UHIi$^S$
hereafter) and AVG (UHIi$^A$ hereafter) gradually increases after mid-afternoon and is the largest at night. This
diurnal pattern is consistent with previous reports in cities exposed to different geographical and climatic
conditions because rural areas cool faster than urban areas (Oke et al., 2017). Additionally, UHIi$^A$ is positive
throughout all days ranging from 0.2–2.2 °C (i.e., warmer than rural area, GP) and is greater than UHIi$^S$ by 0–
1.5 °C. A possible reason for this stronger UHIi$^A$ is that the AVG stations are located in the central business
district; thus, the densities of buildings surrounding these stations are much higher than those surrounding the SFP





station. At night (19:00–06:00), UHIi$^A$ and UHI$^S$ are approximately 1.8 °C and 1.4 °C, respectively. The maximum
UHIi difference between the AVG and SFP was 0.7 °C in 2013 and 0.5 °C in 2014.
Around sunrise, sharp declines in the UHIi are observed because the air temperature near the urban area increases
relatively slowly as urban fabrics, such as asphalt, brick, and concrete, have larger heat capacities and less sky
view factors than the rural areas (Oke et al., 2017). Eventually, this slow increase in the air temperature reduces
the differences in $T_{air}$ among the stations, thereby reducing the UHIi. The minimum UHIi$^A$ values were 0.3 °C
(2013) at 09:30 and 0.2 °C (2014) at 08:30, while the minimum UHIi$^S$ occurs at 10:30 with values of –0.1 °C
(2013) and 0.0 °C (2014). This implies that the timing of the minimum UHIi is delayed in the SFP compared to
the AVG. Our findings indicate that the urban forest has a similar air temperature in the daytime as compared to
the rural area (i.e., GP) where has a lower thermal admittance because of its location within the airport. Especially
when there is strong ET and more time is required to warm the SFP surface, the urban-rural difference in thermal
admittance becomes relatively small. This can be attributed to the higher thermal capacity of the wetter soil of the
SFP as a result of artificial irrigation and the absence of impervious surfaces (Oke et al., 1991).
The diurnal variations in UHIi$^S$ also show the interannual variability in both amplitude and steepness over the two
consecutive years. Despite the similar summertime UHIi$^A$ for both years, the daytime UHIi$^S$ in 2013 was
approximately 0.2 °C lower than that in 2014. Notably, the summer $Q_E$ was greater in 2013 than in 2014, and this
observed summertime asymmetric difference between the SFP and AVG stations was not found in the winter
when ET was negligible (not shown here).
Our results suggest that urban forests can play a significant role in mitigating the thermal environment because of
the wetter soil surface of the park and subsequent increases in $Q_E$, compared to the impervious surfaces in urban
areas. In particular, our findings indicate that the heat mitigation of the urban forest depends on the ratio of $Q_E$ to
net radiation. Indeed, there is an evident negative relationship between daytime $Q_E$ and air temperature differences
between the SFP and AVG stations (Fig. 8). As $K_\downarrow$ is more partitioned to $Q_E$, $T_{air}$ of the SFP decreases more than
that of the AVG, and the maximum temperature difference is observed in the summer season. The SFP is cooler
than the AVG by up to 0.6 °C, but the SFP is warmer than the AVG during the winter-dormant season when ET
is small.
**3.4 Temporal dynamics of net CO₂ exchange**
Figure 9 shows the diurnal evolution of $F_C$ and footprint-weighted road fraction ($\lambda$). Overall, the mean daytime
$F_C$ is negative (i.e., carbon uptake) in the summer (June–August), indicating that photosynthesis, the only carbon
sink, is dominant. This carbon uptake period is coincident with the active vegetation manifested by increases in
EVI (not shown here). Summertime photosynthetic carbon uptake ($GPP$) has a daily average of 7.6 μmol m$^{-2}$ s$^{-1}$
with a maximum of 18.9 μmol m$^{-2}$ s$^{-1}$ around 12:30 (Fig. 8a in Lee et al., 2021). A daily minimum $F_C$ also occurs
around 12:30 with the maximum photosynthetic carbon uptake during this time accordingly. The vegetation
around the SFP absorbs more $CO_2$ than carbon sources and $F_C$ becomes negative only during the summer daytime.
However, because of substantial amounts of anthropogenic emissions and ecosystem respiration, $F_C$ changes from


negative (i.e., carbon sink) to positive values (i.e., carbon source) even around 16:30 in summer unlike in natural
ecosystems, despite the substantial downward shortwave.
$CO_2$ uptake is highest in June, with a maximum of approximately 13 µmol m$^{-2}$ s$^{-1}$ (Fig. 9a). In the middle of
summer (4th and 31st two-week data in Fig. 9a), $CO_2$ uptake decreases significantly because photosynthesis is
limited because of the reduced $K_\downarrow$ by cloud and rainfall with the onset of the summer monsoon (Fig. 2c). This
mid-summer depression of carbon uptake has been reported in the Asian natural vegetations (e.g., Kwon et al.,
2009; Hong and Kim, 2011; Hong et al., 2014). Higher reduction in $CO_2$ uptake observed in 2013 than in 2014
was attributed to a longer monsoon period in 2013. Indeed, from 8 to 21 July 2013 (4th two-week data in Fig. 9a),
the accumulated precipitation was approximately 400 mm for two weeks, and the daily averaged $K_\downarrow$ was only 70
W m$^{-2}$.
As photosynthesis decreases, $F_C$ changed to positive values from November. During the non-growing season (i.e.,
late autumn, winter, and early spring), anthropogenic emissions were also dominant because photosynthesis and
ecosystem respiration decrease with smaller $K_\downarrow$ and lower temperatures. During these periods, $F_C$ had minimum
values at 04:00–05:00 and increases until 15:00–16:00. Therefore, the diurnal variations in $F_C$ mainly followed
the traffic volume (Fig. 4a in Lee et al., 2021), and there also is a clear positive relationship between $F_C$ and $\lambda$
(23rd, 45th, and 47th two-week data in Fig. 9). It is also noteworthy that the peak time of $F_C$ (16:00) is earlier
than the peak time of $\lambda$ (18:00) from December to early March because $E_B$ is the largest at around 15:00–16:00,
indicating that $E_R$ and $E_B$ are the controlling factor of $F_C$ in this period.
With such apparent seasonal $F_C$ variation, it is notable that its variability depends on the spatio-temporal
distribution of $CO_2$ sources and flux source area because the latter covers various land use with changes in wind
direction and atmospheric stability (Fig. 10). In autumn, the main wind direction changed to the north as the
synoptic conditions change particularly (Fig. 3); therefore, $\lambda$ is smaller in autumn compared to other seasons (Fig.
9b). For example, the road fraction was smallest at < 1% from midnight to midday and < 3% during the afternoon
in October and November (11th, 12th, 36th, and 37th two-week data in Fig. 9b). In these periods, the nighttime
$F_C$ showed the lowest value of approximately 2.9 µmol m$^{-2}$ s$^{-1}$, which was attributable to the smallest road fraction,
lower respiration, and minimal heating usage.
In early spring, $\lambda$ was generally larger; thus, $E_R$ played a significant role in $F_C$, and $E_B$ also remained non-zero
until early April, thereby resulting in the largest $F_C$ in this period. With a shutdown of the heating system (i.e.,
zero $E_B$) and the sprouting of leaves in April, there was a sharp decrease in $F_C$ (Fig. 10b and 10c). From December
to March, $CO_2$ emissions increased up to 30 µmol m$^{-2}$ s$^{-1}$ with larger variability in the south–west direction because
of intermittent anthropogenic emissions from the park facility (due to space heating and boiling water), as well as
the relatively increased contribution of vehicles on the road in the western part of the site.
Although the positive $F_C$ in the winter decreased in spring, its magnitude showed directional differences (Fig.
10c). On the eastern side, the mean $F_C$ showed a negative value in May, whereas it remained positive on the
western side (210–270°) until May. Therefore, these findings further indicating the different contributions of



various carbon sources and sinks among the different wind directions. For the wind directions from the north to
the east (0–120°), $F_C$ showed a relatively weaker carbon sink than other directions because of the relatively low
tree fraction in this direction (Fig. 10a and 10c). On the southern side (150–180°) having the highest tree cover
fraction, a maximum carbon uptake about 15 μmol m$^{-2}$ s$^{-1}$ on average was found in June. However, despite the
dense vegetation on the south and west side (120–330°), the $F_C$ magnitude was much smaller than that of other
natural forests. This is related to the anthropogenic emissions from vehicles on the roads which is discussed in
section 3.6.
**3.5 Light use efficiency of biogenic CO$_2$ components**
$F_C$ at the SFP shows a typical light response to the photosynthetically active radiation (PAR) in a way similar to
natural ecosystems in spite of anthropogenic CO$_2$ sources from vehicles (Fig. 11a and 11b). However, this light
response in the urban forest is a distinct contrast with the non-dependent $F_C$ in high-rise high-population
residential areas in Seoul under the same climatic conditions (EP station). Importantly, *GPP, NBE,* and $F_C$ show
different trends on PAR depending on the direction. As stated in Section 2.1.1 and 3.4, the western side has a
higher density of trees as against more grass on the eastern side, and biotic CO$_2$ uptake from the western side is
substantially larger than that on the eastern side. Accordingly, the slope of the light response curve for PAR on
the western side is steeper than on the eastern side. $F_C$ at zero PAR ($F_{C\_0}$) is larger on the western side (9.7 μmol
m$^{-2}$ s$^{-1}$) than on the eastern side (5.1 μmol m$^{-2}$ s$^{-1}$) because of a contribution of $E_R$ from roads on the western side
of the tower.
*NBE* shows a comparable light response to natural vegetation (e.g., Schmid et al., 2003). A rectangular hyperbolic
equation has been used to examine the light response of *NBE* and elucidate the directional differences in carbon
uptake:
$$NBE = -GPP + RE = -\frac{\alpha \cdot GPP_{sat} \cdot PAR}{GPP_{sat} + \alpha \cdot PAR} + RE \qquad (3)$$
where $\alpha$ is the quantum yield efficiency (the initial slope of the light-response curve), $GPP_{sat}$ is the potential rate
of the ecosystem CO$_2$ uptake. $\alpha$ is approximately 0.0651 and 0.0558 μmol CO$_2$ (μmol photon)$^{-1}$ on the western
and eastern sides, respectively. Notably, $\alpha$ on the western side is comparable to the high initial quantum yield in
crops and subtropical forests in East Asia (Hong et al., 2019b; Emmel et al., 2020). Additionally, $GPP_{sat}$ is 30.9
and 12.7 μmol m$^{-2}$ s$^{-1}$ on the western and eastern sides, respectively. In addition, the light saturation points are at
a PAR of 1500 μmol m$^{-2}$ s$^{-1}$ on the eastern side, which occur at a relatively lower PAR than on the western side.
Daytime respiration estimated from equation (3) is 6.7 and 6.3 μmol m$^{-2}$ s$^{-1}$ on the western and eastern sides,
respectively. Because *GPP* is related to PAR, the difference in monthly cumulative *GPP* between the two years
shows a close relationship with the difference in the monthly sunshine duration ($r^2 = 0.75$, not shown here), thereby
suggesting a possible impact of change in the onset of the summer monsoon on urban forests.
The magnitude of *NBE* from the western side is larger than that from the suburban area having about 50%
vegetative fraction in Montreal, Canada (Fig. 7b in Bergeron and Strachan, 2011) and $F_C$ from a highly vegetated



environment of about 67% vegetative fraction in Baltimore, USA (Crawford et al., 2011). Also, *GPP* from the
western side is comparable to the dense forest canopies in subtropical forests in Korea (Hong et al., 2019b),
deciduous forest ecosystems (Goulden et al., 1996), and a mixed hardwood forest ecosystem (Schmid et al., 2000).
However, *NBE* from the eastern side is similar to $F_C$ from the suburban areas of about 44%, 50%, and 64%
vegetative fraction in Swindon, UK (Ward et al., 2013) and Montreal, Canada (Bergeron and Strachan, 2011), and
Ochang, Korea in the same climate zone (Hong et al., 2019b), respectively.

**3.6 Annual budget of CO₂ sources and sink**

The annual budget of the $F_C$ and its components is summarized in Table 3. The annual sums of the *GPP* and *RE*
in the SFP are 4.6 kg $CO_2$ m⁻² year⁻¹ (1244 gC m⁻² year⁻¹) and 5.1 kg $CO_2$ m⁻² year⁻¹ (1378 gC m⁻² year⁻¹),
respectively. This photosynthetic carbon uptake is smaller than its global mean *GPP* in natural deciduous
broadleaf forests with similar annual precipitation and annual mean air temperature (total 8 years of data from 4
sites of FLUXNET2015 dataset reported in Pastorello et al., 2020) and similar to that of deciduous broadleaf
forests in East Asia (Awal et al., 2010; Kwon et al., 2010) (Table 4). Our speculation is, however, that this GPP
is relatively larger if we consider the low vegetation fraction and leaf area index (LAI) at our urban park. Indeed,
*GPP* is comparable to values reported in other urban sites if it is scaled with the vegetation cover fraction. Previous
studies have shown that the *GPP* of urban vegetation is scaled with vegetation cover fraction with an increase of
about 0.7 kg $CO_2$ m⁻² year⁻¹ per 10% increase in vegetation cover fraction (Awal et al., 2010; Crawford and
Christen, 2015; Velasco et al., 2016; Menzer and McFadden, 2017). GPP at the SFP with a 46.6% vegetation
cover fraction is approximately 1.5 kg $CO_2$ m⁻² year⁻¹ larger than this scale (Fig. 12a).
Eventually, this large *GPP* results in a substantial decrease in $F_C$ when they are scaled by vegetation fraction.
Hong et al. (2019b) reported a linear decrease in *Fc* of approximately 3.0 kg $CO_2$ m⁻² year⁻¹ per 10% increase in
vegetation cover fraction based on the observed $F_C$ across an urbanization gradient in Korea (Fig. 12b). The annual
$F_C$ in the SFP of 7.1 kg $CO_2$ m⁻² is 1.2 kg $CO_2$ m⁻² year⁻¹ smaller than this scaled relationship (i.e., more carbon
uptake). In particular, $F_C$ in the SFP is approximately 3.0 kg $CO_2$ m⁻² year⁻¹ less than that in recently developed
high-rise high-population urban areas in Seoul. Our results suggest that efficient management of urban forests,
such as regular irrigation and fertilization, can be an efficient way to adapt and mitigate climate change by
increasing CO₂ uptake in artificial forest constructions in East Asia.
Meanwhile, *RE* at our site is much larger than that in temperate deciduous forests in East Asia (Takanashi et al.,
2005; Kwon et al., 2010) and similar to that in the urban forest in East Asia (Awal et al., 2010), as well as to the
global mean *RE* over forests with similar annual precipitation and annual mean air temperatures (Pastorello et al.,
2020). Put differently, the urban forest considered in our study is an outlier compared to other natural forest
canopies and urban forests because *RE/GPP* > 1 (Table 4). Autotrophic respiration is considered to be
approximately half of GPP as a rule of thumb (Piao et al., 2010), which corresponds to approximately 45% of the
*RE* at our site, thereby indicating a large contribution of heterotrophic respiration to *RE*. Indeed, it was reported
that soil respiration at the same site was approximately 4 kg $CO_2$ m⁻² year⁻¹ (Bae and Ryu, 2017). The reason for
the large soil organic carbon was mainly because rice cultivation was carried out in this region before the 1950s,





and organic carbon-rich soil was transplanted during the SFP construction, and fertilizers were applied regularly.
It has also been reported that $RE$ is enhanced in urban areas because of the relatively warmer temperature in urban
regions (i.e., UHI) (Awal et al., 2010). Notably, $Q_{10}$ (the rate by which respiration is multiplied when temperature
increases by 10 °C) is about 1.9 at the site and matches the $Q_{10}$ value for ecosystem respiration (2.2 ± 0.7)
calculated for natural forests across 42 FLUXNET sites (Mahecha et al., 2010). Further analysis based on the
observed $Q_{10}$ and the UHIi at the SFP indicates that UHI leads to an approximately 5% increase in $RE$.
Figure 13 shows the monthly cumulative sum of the $F_C$ and its partitioned components. Seasonal variations in the
strength of carbon sources and sink as well as $F_C$ are mainly regulated by the biogenic component in summer and
the anthropogenic component in winter. Furthermore, $F_C$ is minimum in June, despite the similar $GPP$ from June
to August because of the relatively smaller $RE$ during the summer season. Even in summer, photosynthetic carbon
uptake is balanced with ecosystem respiration and does not offset all biotic and anthropogenic emissions, thus
resulting in positive $F_C$ values throughout the year. In winter, $E_B$ is dominant with negligible $GPP$ and $RE$ due to
cold temperatures, and $E_R$ also becomes larger than $RE$ from November. $E_R$ shows apparent seasonal variation in
wind direction and atmospheric stability. Its magnitude is about 0.0666 $\mu$mol m$^{-2}$ veh$^{-1}$ h$^{-1}$ in neutral condition
and consistent with the value in the inventory data (Lee et al., 2021). The average monthly traffic speed for the
road in front of the SFP is 50–60 km h$^{-1}$ (based on the January 2014 data from the Seoul Metropolitan Government
Traffic Speed Report), and the $CO_2$ emission rate is approximately 150 g $CO_2$ km$^{-1}$ veh$^{-1}$ based on the emission
data at this speed (Kim et al., 2011). With the width of the ten-lane road (25–30 m), the inventory-based slope
(i.e., $CO_2$ emission rate per vehicle per area per half-hour) is approximately in the range of 0.0631–0.0757 $\mu$mol
m$^{-2}$ veh$^{-1}$ half-hour s$^{-1}$ ($\cong$ 150 gCO$_2$ km$^{-1}$ veh$^{-1}$ × 1/30 or 1/25 m$^{-1}$ × 1/44 mol gCO$_2$$^{-1}$× 10$^{-3}$ km m$^{-1}$ × 10$^6$ $\mu$mol
mol$^{-1}$ × 1/1800 half-hour s$^{-1}$).
There is an evident yearly difference in individual carbon sources and sink in two consecutive years. $E_B$ is mainly
caused by heating buildings and hot water in park facilities using natural gas. Notably, $E_B$ is also smaller in the
first year because of the relatively smaller number of park visitors and consequently smaller gas consumption,
compared to the second year. Indeed, $E_B$ is highly correlated with gas consumption in SFP during winter on a
monthly basis ($R^2$ = 0.94; Fig. 6 in Lee et al., 2021). Eventually, these annual differences lead to a smaller annual
mean total $F_C$ in the first year than in the second year (Table 3). However, $RE$ is maximum in the August of the
first year, while it is highest in July of the second year because of the interannual variations in air temperature
with changes in the timing and duration of the East Asian summer monsoon, of which impacts have also been
reported in natural vegetation in the same region (Hong and Kim, 2011; Hong et al., 2019b). In other words, the
monthly mean air temperature is highest in August of the first and July of the second year because of the short
East Asian monsoon period and drought in July of the second year. However, the $GPP$ in summer is relatively
smaller in the first year by the mid-summer depression of solar radiation because of the elongated monsoon period
(Fig. 2). However, $GPP$ does not shrink in the second year of significant drought because there is ample water
supply by a sprinkler. Our results emphasize the important role of forest management in enhancing carbon uptake
and evaporative cooling despite the low vegetation fraction.



### 4 Summary and conclusions

This study reported two-year surface fluxes of energy and $CO_2$ measured by the eddy covariance method while also examining the role of artificially generated urban forests in mitigating air temperature and anthropogenic $CO_2$ emissions. The study area is located in the East Asian monsoon region, characterized by a lengthy summer rainy season. During the measurement period, the second year was contrasted with the first year because of the drought compared to the normal climate condition in the first year. The study region is a park with an artificially planted forest in the Seoul Metropolitan Area. The urban forest had a heavy traffic volume around it and was redeveloped from a racetrack and factory in the mid-2000s. To examine the mitigation of air temperature, this study compared meteorological conditions in the urban forest with the surrounding high-rise high-population urban areas. This study also proposed a statistical $CO_2$ flux partitioning method based on temporal subsets of flux data and high-resolution footprint-weighted land use data to understand the abiotic and biotic contributions to $F_C$.

Surface energy balance in the SFP is influenced by the summer monsoon, and more energy is distributed to $Q_E$ than $Q_H$ in the summer when vegetation is active, similar to natural forests in this climate zone. Therefore, the Bowen ratio in this urban forest ranges from near 0 (summer) to about 4 (winter), which is lower throughout the year than that of high-rise and high-density residential areas in Seoul. This suggests that the vegetation and unpaved surfaces of urban forests facilitate more evaporative cooling compared to the impervious surfaces in urban areas. Furthermore, ET decreased in the second year when there was a drought, but this drop was not as much as the reduced precipitation if we consider the substantial changes in precipitation and radiative forcing in two consecutive years.

It is also evident that the urban forest reduced the warming trend and UHIi around the study area. Air temperature in the SFP was lower than the surrounding area, but this coolness was reinforced after the park was created. The warming trend diminished after the construction of the park and was smaller than that in other urban regions in the Seoul Metropolitan Area. In addition, the construction of the park delayed the timing of the maximum temperature difference between the urban forest and high-rise commercial from the morning to the afternoon, coinciding with the timing of the maximum $Q_E$. The SFP shows a general diurnal UHIi variation pattern, which has a higher temperature at night than in rural areas. However, the UHIi in SFP is lower by 0.6 °C in summer compared to the surrounding urban area, and the minimum peak time is delayed, possibly because vegetation and permeable soils in SFP have a larger thermal capacity. Notably, UHIi decreased more in the partitioning of incoming energy into latent heat fluxes. As a rule of thumb, there was cooling by 0.2 °C compared to the surrounding urban area if $Q_E/K_\downarrow$ increased by 10%.

Net $CO_2$ exchange at the urban forest showed typical temporal variations in natural forest canopies influenced by the East Asian summer monsoon (Hong and Kim, 2011; Hong et al., 2019b). A mid-summer depression of carbon uptake was observed with the onset of the summer monsoon, like vegetation in the East Asian monsoon region. The $GPP$ was estimated by the statistical partitioning method, and the non-zero $GPP$ period was coincident with the active vegetation of the significant vegetation index. Summertime photosynthetic carbon uptake had a daily average of 7.6 $\mu$mol m$^{-2}$ s$^{-1}$ with a maximum of 18.9 $\mu$mol m$^{-2}$ s$^{-1}$ around 12:30. However, even during the growing





season, vegetative carbon uptake was insufficient to offset anthropogenic $CO_2$ emissions and ecosystem respiration on a time scale of > 1 day. Our estimations of anthropogenic $CO_2$ emissions from vehicles and buildings agreed with the estimations based on inventory data such as $CO_2$ emission rate of vehicles and monthly gas consumption, and their annual budgets each had a comparable magnitude to $GPP$.

Annual $GPP$ of the urban forest was relatively smaller than that of the forest in East Asia exposed to similar climatic conditions because of the relatively smaller vegetation cover fraction and LAI. However, it was larger than the $GPP$ expected from the relationship from previous urban studies if it was normalized by the vegetation cover fraction. $RE$ is, however, much larger than that in the temperate East Asian forests and similar to the urban forest in East Asia. We speculate that soil respiration enhanced such large ecosystem respiration by relatively warmer temperatures in a city and rich soil organic carbon in the SFP. Eventually, the annual mean total $F_C$ is 7.1 kg $CO_2$ m$^{-2}$ year$^{-1}$, which is smaller than the estimate from the scaling between annual total $F_C$ and vegetation fraction (Hong et al., 2019b). Because of the spatial heterogeneity, $F_C$ and its components showed directional changes. $NBE$ from the eastern side is similar to $F_C$ of suburban areas with approximately 44%, 50%, and 64% vegetative fraction in Swindon, UK (Ward et al., 2013) and Montreal, Canada (Bergeron and Strachan, 2011), and Ochang, Korea in the same climate zone (Hong et al., 2019b), respectively. However, the $NBE$ and $GPP$ from the western side are comparable to dense forest canopies in subtropical forests in Korea (Hong et al., 2019b), deciduous forest ecosystems (Goulden et al., 1996), and a mixed hardwood forest ecosystem (Schmid et al., 2000).

Our study reveals that urban forests make significant traces of air temperature and $CO_2$ fluxes despite their relatively small area. Our key findings are that urban forests in East Asia are highly influenced by the East Asian monsoon like natural forests in this region, but such influence is mitigated by artificial irrigation and fertilization in urban forests. In particular, our results emphasize the importance of forest management for efficient carbon uptake and evaporative cooling despite the low vegetation fraction. Furthermore, our observation study also indicates that caution in soil management is necessary to reduce $CO_2$ emissions in urban forests, mainly resulting from large soil organic carbon. We also highlight that our statistical $CO_2$ flux partitioning is a promising method to improve our understanding of the carbon cycle in urban and suburban areas, and a more extensive study is required for validation in another geographical zone.

*Acknowledgment*. This research was supported by the Korea Meteorological Administration Research and Development Program under Grant KMI2021-01610 and National Research Foundation of Korea Grant from the Korean Government (MSIT) (NRF-2018R1A5A1024958). All data and codes are available in Lee et al. (2021) and upon request to the corresponding author (jhong@yonsei.ac.kr / https://eapl.yonsei.ac.kr).



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



Table 1. Details of the stations used in this study.

| Sites | Location | LCZ | Height [m] | Used variables |
|---|---|---|---|---|
| **Eddy covariance station** | | | | |
| SFP (Seoul Forest Park) | 37.5446°N, 127.0379°E | $LCZ_A$ | 12.2 | |
| EP (Eunpyeong) | 37.6350°N, 126.9287°E | $LCZ_1$ | 30 | *Flux* |
| **Automatic weather station** | | | | |
| SD (Seongdong) | 37.5472°N, 127.0389°E | $LCZ_{5B}$ | 25 | *$T_{air}$, RH, WS, WD, Precipitation* |
| AVG | | | | |
| (Gangnam) | 37.5134°N, 127.0467°E | $LCZ_{21}$ | 59 | |
| (Seocho) | 37.4889°N, 127.0156°E | $LCZ_{21}$ | 35.5 | *$T_{air}$* |
| (Songpa) | 37.5115°N, 127.0967°E | $LCZ_{15}$ | 58.2 | |
| **Aerodrome meteorological observation station** | | | | |
| GP (Gimpo) | 37.5722°N, 126.7751°E | $LCZ_D$ | 11.4 | *$T_{air}$* |





Table 2. Gap-filled annual budgets for surface energy fluxes and precipitation (P).

| Sites | $ET$ (mm) | $Q_H$ (MJ m$^{-2}$) | $Q_E$ (MJ m$^{-2}$) | $Q^*$ (MJ m$^{-2}$) | P (mm) |
|---|---|---|---|---|---|
| 1st year (2013.06 – 2014.05) | 367 | 726 | 896 | 1797 | 1256 |
| 2nd year (2014.06 – 2015.05) | 320 | 867 | 781 | 1848 | 932 |
| Mean annual sum of two-year | 344 | 797 | 839 | 1823 | 1094 |







Table 3. Gap-filled annual budgets for $F_C$ (observed by EC measurement) and its components, indicating ecosystem respiration ($RE$), photosynthetic uptake by vegetation ($GPP$), vehicle emissions ($E_R$), and building emissions ($E_B$). All fluxes are in kg $CO_2$ m$^{-2}$ year$^{-1}$.

| Sites | $F_C$ | $RE$ | $GPP$ | $E_R$ | $E_B$ |
|---|---|---|---|---|---|
| 1$^{st}$ year (2013.06 – 2014.05) | 6.6 | 5.1 (71%) | 4.7 (64%) | 3.3 (76%) | 1.0 (20%) |
| 2$^{nd}$ year (2014.06 – 2015.05) | 7.6 | 5.0 (77%) | 4.5 (70%) | 3.2 (81%) | 1.9 (15%) |
| Mean annual sum of two-year | 7.1 | 5.1 (65%) | 4.6 (59%) | 3.3 (71%) | 1.5 (25%) |



Table 4. Annual budgets of biogenic $F_C$ components and ratios in deciduous broadleaf forests in similar climatic conditions reported in previous studies. All fluxes are in kg $CO_2$ m$^{-2}$ year$^{-1}$.

| Sites name | Reference | MAT (°C) | MAP (mm) | maximum LAI | RE | GPP | NBE | RE/GPP |
|---|---|---|---|---|---|---|---|---|
| Seoul Forest Park | This study | 13.9 | 1094 | 1.6 | 5.1 | 4.6 | +0.5 | 1.11 |
| Nagoya urban forest | Awal et al. (2010) | 15.9 | 1680 | 5.5 | 4.9 | 6.2 | −1.3 | 0.74 |
| Toyota rural forest | | 14.5 | 1518 | 4.5 | 2.6 | 4.6 | −2.0 | 0.56 |
| Gwangneung deciduous forest | Kwon et al. (2010) | 12.8 | 1487 | 5 | 3.8 | 4.1 | −0.3 | 0.93 |
| Kiryu Experimental Watershed | Takanashi et al. (2005) | 14.1 | 1309 | 5.5 | 3.9 | 5.6 | −1.7 | 0.70 |
| FLUXNET2015 dataset* | Pastorello et al. (2020) | 14.5 | 1113 | | 4.1 | 6.0 | −1.9 | 0.68 |

*Average value of 8-year data from 4 sites having mean annual temperature (MAT) of 12-16°C, mean annual precipitation (MAP) of 900-2000 mm.



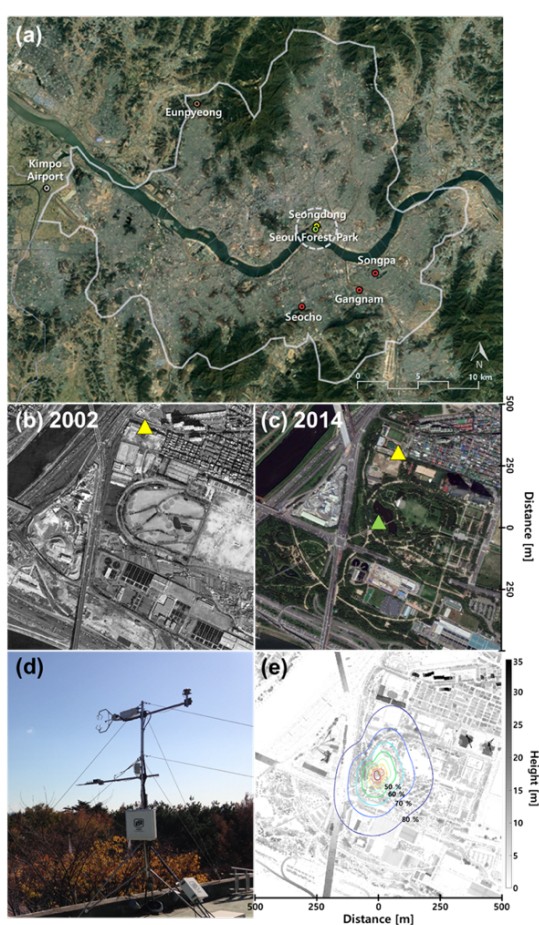

783

Figure 1. Site descriptions. (a) Location of the stations in Seoul (modified from map data © Google Earth 2019),
(b) aerial photographs around Seoul Forest Park (SFP) in 2002 before the creation of the park and (c) in 2014
during the observation period (SFP; *green triangle*, SD; *yellow triangle*), (d) photograph of the SFP station, and
(e) footprint climatology (Hsieh et al., 2000) with the height of surrounding obstacles around the SFP station.


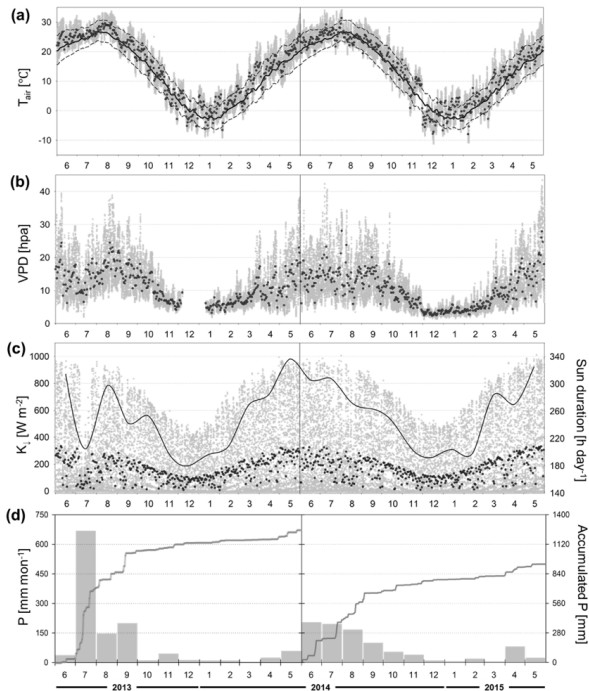

Figure 2. Climatic conditions of the SFP for two years from June 2013 to May 2015: 30-min (*gray dots)* and daily
mean (*black dots*) (a) air temperature with 30-year normal values of Seoul (daily mean; *solid line*, min and max;
*dashed lines*), (b) vapor pressure deficit (VPD) and missing data existing on December 2013, (c) downward
shortwave radiation ($K_\downarrow$) and monthly averaged sunshine duration per day (*black line*), (d) monthly precipitation
(*gray bars*) and yearly accumulated precipitation (*solid line*).



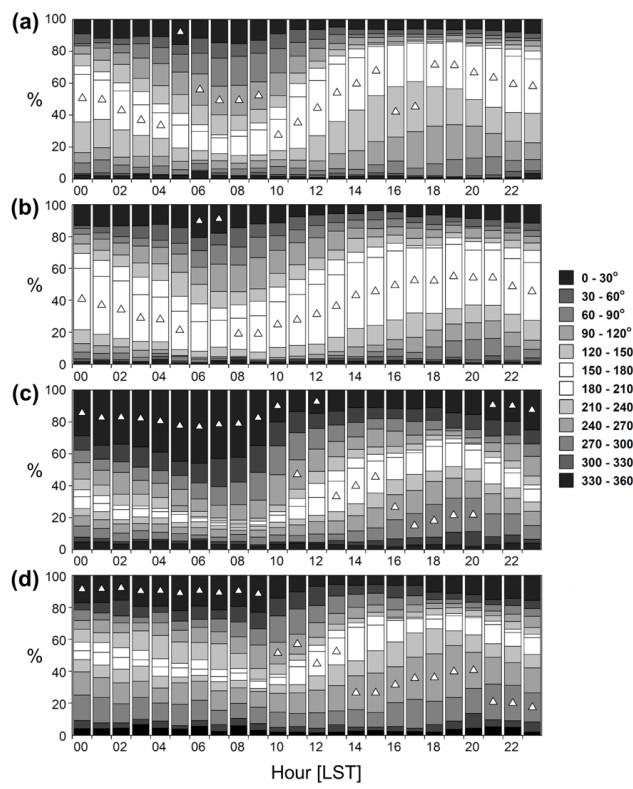

Figure 3. Seasonal mean diurnal courses of wind direction for 30° intervals. Each column represents the hourly

ratio of the wind direction of the season: (a) spring (b) summer (c) autumn (d) winter. The white triangle indicates

the dominant wind direction during that hour.





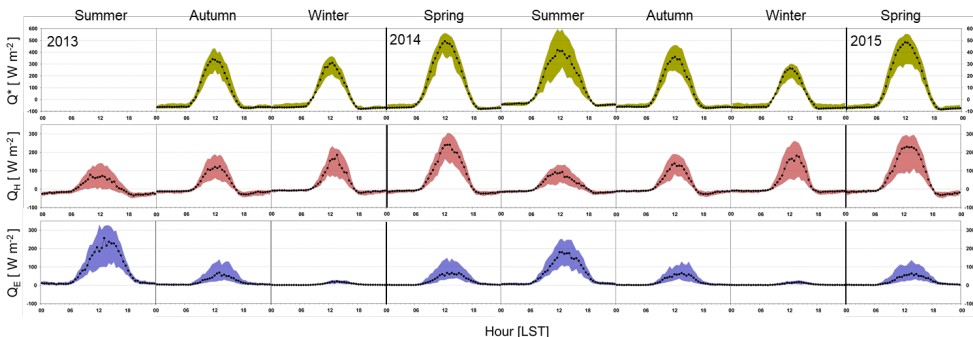


Figure 4. Diurnal variations of surface energy fluxes. Seasonal median diurnal variations (*points*) and interquartile
ranges (*shaded*) of 30-min sensible heat flux ($Q_H$), latent heat flux ($Q_E$), and net radiation ($Q^*$) for two years.
Since the net radiation system was installed in September 2013, there was no $Q^*$ value in the first summer.





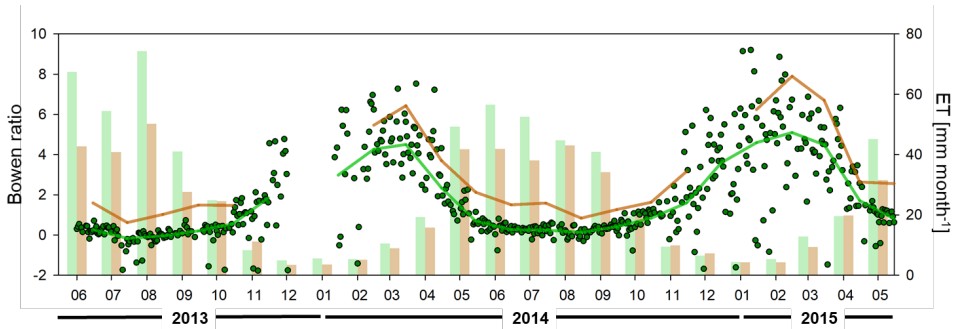


Figure 5. Daily Bowen ratio ($\beta = \sum Q_H / \sum Q_E$; *dots*), monthly Bowen ratio *(lines)*, and gap-filled monthly

evapotranspiration (ET; *bars*) for two years (SFP; *green*, EP; *brown*).



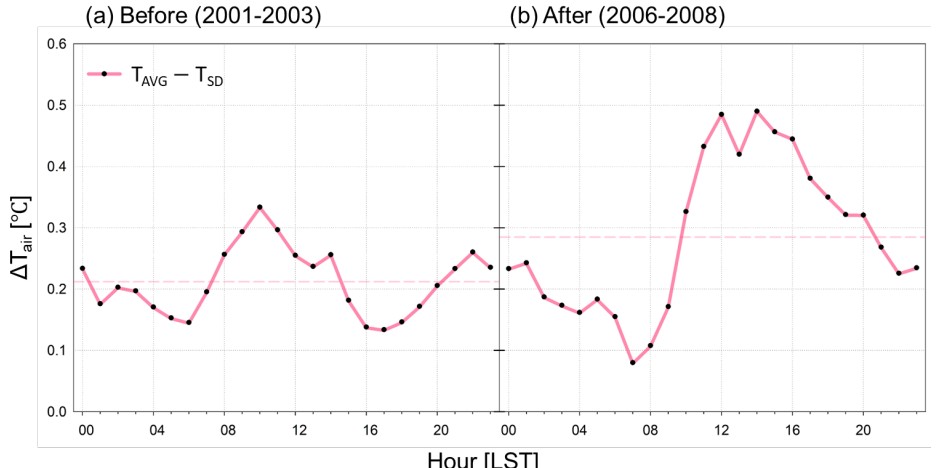


Figure 6. Mean diurnal pattern of air temperature difference ($\Delta T_{air}$) between AVG and SD (a) before and (b) after
the construction of the park in summer. AVG indicates an average of three automatic weather stations (Gangnam,
Seocho, Songpa) in Seoul. The red dash line indicates the mean $\Delta T_{air}$ before and after the construction of the park.

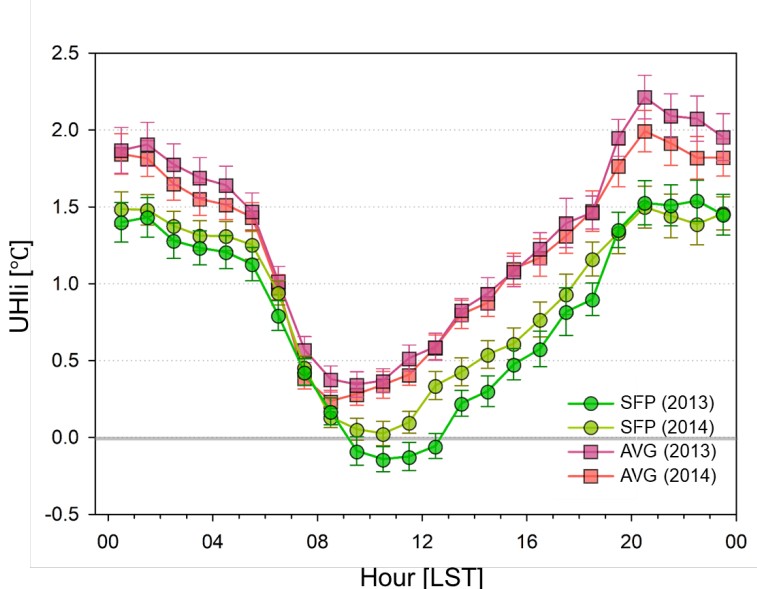

Figure 7. Hourly mean diurnal variation of the urban heat island intensity (UHIi) of the SFP and AVG in the
summer of 2013 and 2014. The error bars represent standard errors.






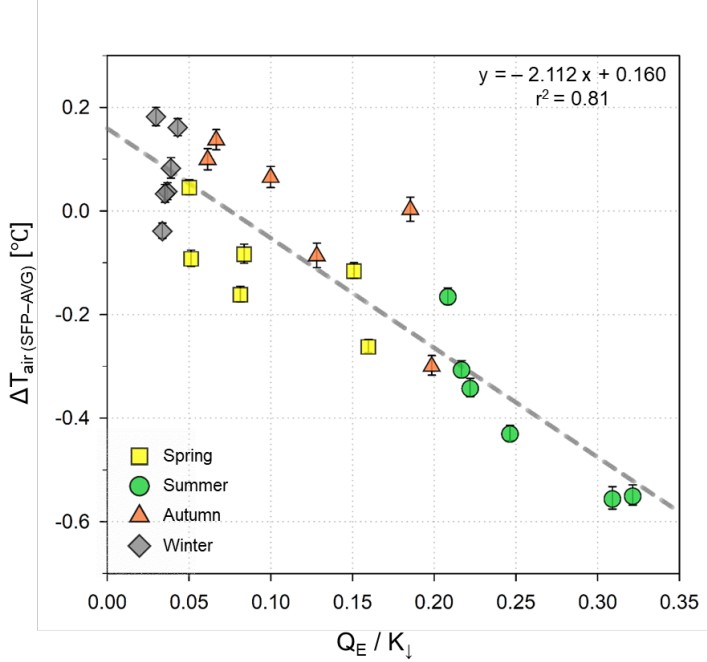


Figure 8. Relationship between the ratio of monthly $Q_E$ to $K_\downarrow$ and mean air temperature difference between SFP
and AVG during the daytime ($K_\downarrow$ > 120 W m$^{-2}$) for two years. The error bars represent standard errors.

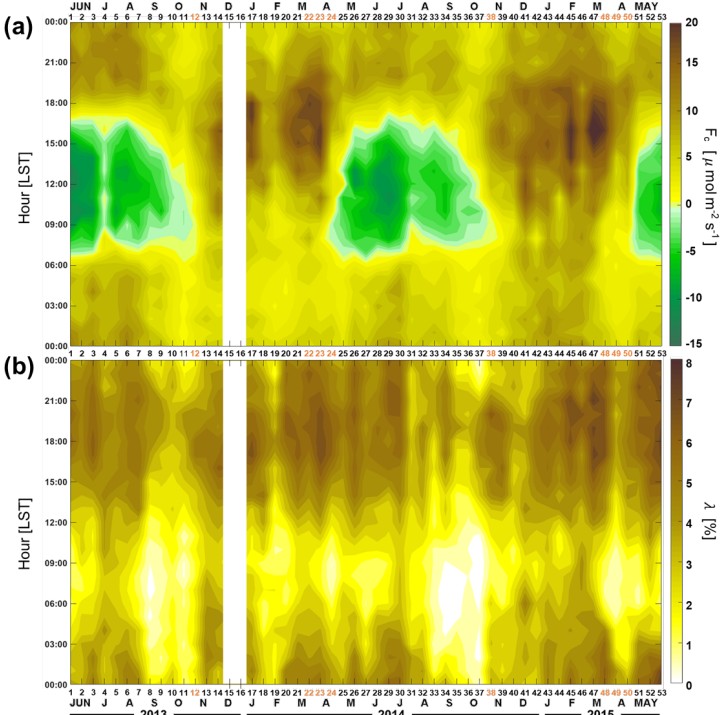

Figure 9. (a) Temporal variation of hourly averaged $F_C$ and (b) source area weighted road fraction ($\lambda$) for every two-week. The horizontal axis indicates the order of every two-week for two years, and the vertical axis is the time of day. In December 2013, there was a gap for approximately 4 weeks due to the power system failure. The yellow numbers indicate the two-week (12[th], 22[nd]–24[th], 38[th], and 48[th] –50[th]) having the transition period when the observed $F_C$ is primarily attributable to traffic emissions ($E_R$).



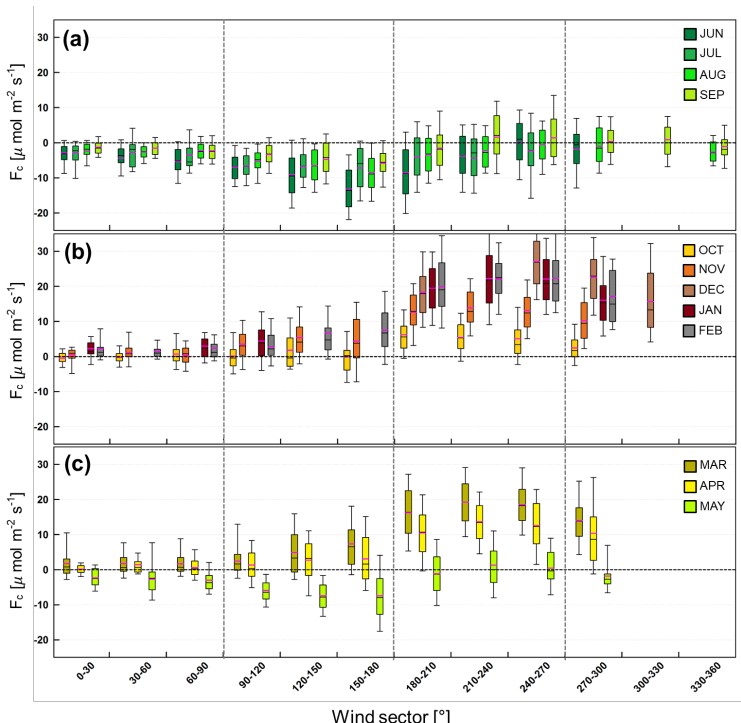

Figure 10. Monthly boxplots of daytime ($K_\downarrow > 120$ W m$^{-2}$) $F_C$ by wind direction. Boxes have a minimum of 20 samples. Box limits are upper and lower quartiles, and whiskers are distances of 1.5 times the interquartile range from each quartile. Median and mean values are indicated by the black and pink horizontal lines.



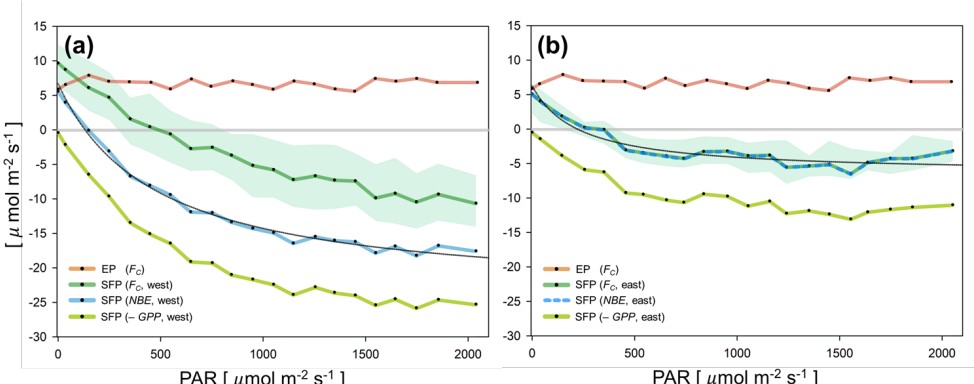

Figure 11. During the growing season (June–August 2013, 2014), light-response curves as a function of photosynthetically active radiation (PAR, in bins of 100 $\mu$mol m$^{-2}$ s$^{-1}$): (a) for the western sectors (150° < Φ < 300°) and (b) for the eastern sectors (30° < Φ < 90°). Black line is a rectangular hyperbolic equation fitting net biome exchange ($NBE = RE - GPP = F_C - E_R$) to PAR, and EP (*brown line*) is a light-response curve for the high-rise high-population residential area in Seoul. The shaded areas indicate interquartile range.





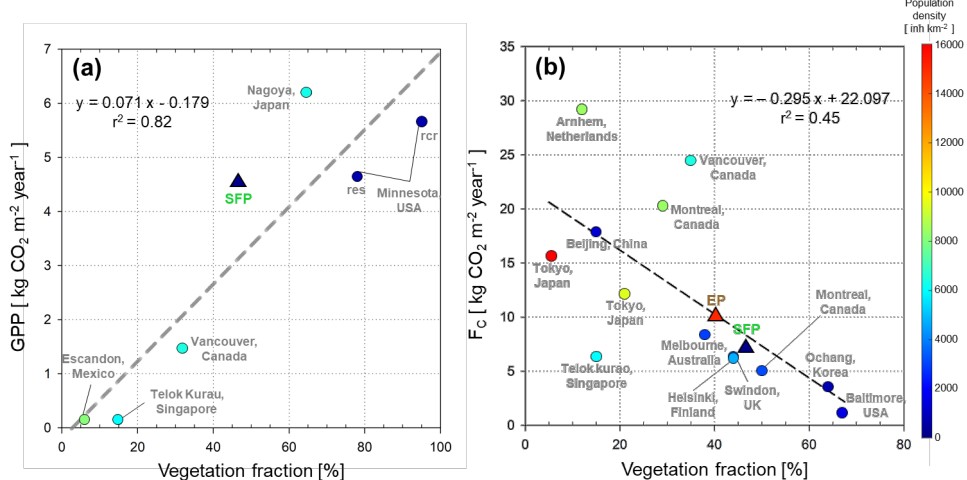


Figure 12. Relationship between vegetation fraction (a) annual *GPP* and (b) annual $F_C$ in urban sites (Fig. 12a in
Hong et al., 2019b).






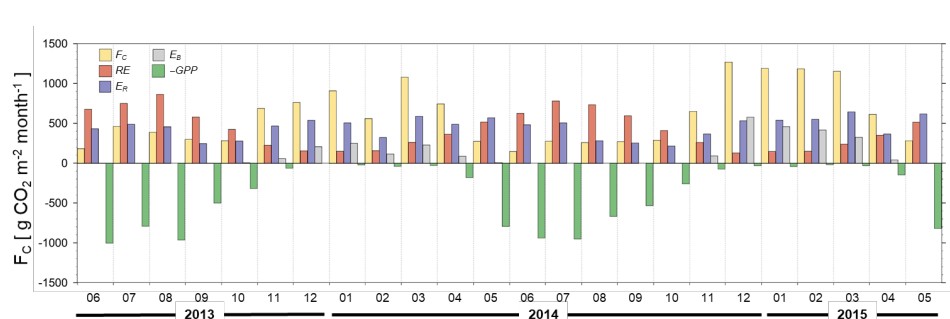


Figure 13. Monthly sums for gap-filled $F_C$ (*yellow bar*) with *RE* (*red bar*), $E_R$ (*blue bar*), $E_B$ (*gray bar*), and –
*GPP* (*green bar*)