# Peer review of "Traces of urban forest in temperature and CO2 signals in monsoon East Asia"

_Atmospheric Chemistry and Physics, 2021_

## Author Comment (AC1)

**Review of ACP-2021-354**

This paper presents energy and CO2 exchange at a complex urban forest site in Seoul, Korea. Sub-daily, seasonal and interannual differences in the observed fluxes are analysed and related to climate conditions (e.g. the effect of the monsoon, drought) and characteristics of the site (e.g. land cover, the proportion of road in the source area). Urban-rural differences in CO2 exchange and the role of the urban forest in reducing the urban heat island are discussed. The observed net CO2 flux is analysed in terms of its anthropogenic and biotic components, although the method of partitioning is not described in this paper.

Overall, this is an interesting and useful study. There have been relatively few studies of CO2 exchange made in urban forests (with most similar studies focusing either on non-urban forests or on more built-up urban neighbourhoods). The detailed analysis at this complex site takes into consideration numerous potential factors which could affect the measurements. The comparison with a more built-up urban site nearby is very interesting and demonstrates the impact of surface characteristics on energy partitioning. Assessing two main roles of urban vegetation (i.e. helping to offset CO2 emissions and cool the urban environment) as well as the effect of land management and weather conditions is also relevant for urban planning and broadens our knowledge of interactions in urban climates.

Generally the paper is in good shape (clearly written, well-structured, suitable figures). My main concern is that there is not enough information given about the partitioning method used. Although the method (and some of the analysis) is described in a MethodsX article that is often referenced here, to make this a standalone publication more of the important information needs to be readily available to readers of this paper. This also makes the review challenging, since it is difficult to assess the suitability of the approaches and validity of the findings without really knowing how the data have been treated. I have several other queries and suggestions about various aspects - please see below for detailed comments.

**Reply:** Thank you for your constructive and critical review. We tried to revise our manuscript based on the reviewer's comments without substantial overlaps with our previous publications of Kent et al. (2018), Hong et al. (2020), and Lee et al. (2021).

Kent, C. W., Lee, K., Ward, H. C., Hong, J. W., Hong, J., Gatey, D., and Grimmond, S.: Aerodynamic roughness variation with vegetation: analysis in a suburban neighbourhood and a city park. Urban Ecosystems, 21(2), 227-243, 2018.

Hong, J. W., Lee, S. D., Lee, K., and Hong, J.: Seasonal variations in the surface energy and CO2 flux over a high-rise, high-population, residential urban area in the East Asian monsoon region.

International Journal of Climatology, https://doi.org/10.1002/joc.6463, 2020.

Lee, K., Hong, J. W., Kim, J., and Hong, J.: Partitioning of net CO2 exchanges at the city-atmosphere interface into biotic and abiotic components. MethodsX, 8, 101231, 2021.

**Major issues**

Methods described elsewhere

L210-1 As the partitioning of Fc is central to some of the results, more information is needed here for this paper to be a standalone publication. A paragraph should be added which summarises the approach used and highlights any important caveats. This will help the reader to understand roughly what has been done to the data (they can still read the MethodsX article for the details) so that they can interpret the results of this study.

**Reply:** As the reviewer suggested, we revised our manuscript by providing more information on the partitioning method.

There should also be a paragraph which gives an overview of how the gap-filling of Fc was done. This is not an easy task in urban areas with many different controls and a heterogeneous source area. Sufficient information needs to be given here in order for the reader to understand what the annual sums given in Section 3.6 mean.

**Reply:** As the reviewer suggested, we revised our manuscript by incorporating the gap filling method.

Measurement height needs further consideration

L123-136 The site characteristics described here are difficult to follow. Different distances from the tower are used and several values are given for the roughness length and displacement height without any clear justification for the values chosen. In L195-198 the measurement height is justified based on a relation including displacement height, but it is not clear what displacement height was used to decide this and for some of the values given in Section 2.2.1 this relation is not satisfied. By other commonly used relations (e.g. a measurement height of at least twice the mean obstacle height), the tower would be too low. Careful explanation is needed here so the reader can clearly follow the justification. Since the measurement height is fairly low, the authors may consider adding additional information to convince the reader. For example, did spectra or turbulence characteristics

(e.g. the drag coefficient) suggest the measurements were sufficiently high, or problematic for certain wind sectors? If there is a strong possibility the measurements were made lower than typically recommended, is it possible to say whether this would affect the findings of the paper?

**Reply:** We estimated aerodynamic roughness parameters using the 1-m horizontal resolution land cover data and should report their values with different wind direction because of different heights of surface bluffs. Please consider that detailed analysis on roughness parameters and integral turbulence characteristics at the site were already reported in Kent et al. (2018). Furthermore, the blending height is pretty low in the skimming flow region because building fraction is less than about 2% within a tower footprint. For better readability, we revised the texts to show roughness length and displacement heights with different wind direction so that we can check the height criteria. We also want to mention that eddy-covariance method is only way to measure turbulent fluxes even in the roughness sublayer over forest canopies and the main footprint covered the forest canopies although lower height may be concern over heterogeneous sources in urban building canyon where there are substantially heterogeneous source and sink distributions.

**Minor issues**

L26 It's not clear here whether the urban population will increase by 68% by 2050 or whether the urban population will reach 68% by 2050
**Reply:** We revised the text as the reviewer suggested.

L33-4 Delete 'as opposed to gray spaces' as it is not needed and does not really fit here (gray spaces are also exposed to the range of conditions mentioned)
**Reply:** We revised the text as the reviewer suggested.

L37 Some examples of harmful effects would be helpful
**Reply:** We revised the text as the reviewer suggested.

L46-7 May be helpful to explicitly mention air-conditioning here if that is what is implied
**Reply:** We revised the text as the reviewer suggested.

L51-2 (and other places) What is meant by direct heat fluxes? Surface heat fluxes? Turbulent heat fluxes? Sensible heat flux?
**Reply:** We revised the text to sensible heat fluxes.

L92 I did not find Appendix A

**Reply:** We are sorry that appendix was missed, and we added appendix A in our revised manuscript. Thank you for your careful check.

L100-1 What about contributions to respiration from visitors to the park?

**Reply:** Human respiration is about 0.4 μmol m$^2$ s$^{-1}$ at most based on the number and staying of park visitors and park areas. We revised the text by adding this information.

L101-3 This sentence doesn't really fit here – merge with the Introduction, Section 3.4, or delete

**Reply:** We revised the text as the reviewer suggested.

L110-2 As these factors are central to the paper, consider adding some more examples here, such as transport options, fuel types, heating demand, weather conditions, etc

**Reply:** We revised the text as the reviewer suggested.

L111 The location of the tower does not affect the anthropogenic emissions, but rather the measured fluxes

**Reply:** We revised the text as the reviewer suggested.

L180 Are these really 'observatories', or would it be more accurate to say 'weather stations'?

**Reply:** We revised the text as the reviewer suggested.

L204-5 Please add a sentence to justify why negative CO2 fluxes were discarded during nighttime

**Reply:** Please make sure that vegetative photosynthesis is an only sink and accordingly, negative fluxes during the nighttime are not physically feasible because there are no sinks of $CO_2$ at night. For better readability, we added the text into the manuscript.

L214 (and elsewhere) the surface energy balance is often mentioned but only net radiation and the

turbulent heat fluxes are analysed. At least the contributions of the storage heat flux (various components) and anthropogenic heat flux need to be considered for an energy balance study. If neither of these were estimated, the discussion should refer to the radiation and turbulent fluxes (and not the surface energy balance)

**Reply:** We revised the manuscript as the reviewer suggested.

L218 Is 'sensible heat flux' intended rather than 'surface heat fluxes'?

**Reply:** All turbulent fluxes show mid-summer depression with the seasonal march of the summer monsoon and its related reduction of downward shortwave radiation. This has been reported in many East Asian ecosystems (e.g., Kwon et al., 2009; Hong et al., 2011).

Kwon, H., Park, T. Y., Hong, J., Lim, J. H., and Kim, J.: Seasonality of Net Ecosystem Carbon Exchange in Two Major Plant Functional Types in Korea. Asia-Pacific Journal of Atmospheric Sciences, 45(2), 149-163, 2009.

Hong, J., and Kim, J.: Impact of the Asian monsoon climate on ecosystem carbon and water exchanges: a wavelet analysis and its ecosystem modeling implications. Global Change Biology, 17(5), 1900-1916, 2011.

L231 Should this be Fig 4 not Fig 6?

**Reply:** We revised the text as the reviewer suggested.

L230-5 Not clear here which are new findings from this analysis and which are being referred to in the Hong and Kim (2011) paper

**Reply:** Ecosystem in the East Asia shows unique characteristics in their SEB and photosynthetic carbon uptake with the seasonal march of the East Asian monsoon, which include mid-summer depression of surface fluxes and its interannual variability with changes in intensity and timing of heavy rainy season. Our study is the first report this mid-summer depression in the artificially generated forest. Importantly, unlikely to natural forest in East Asia, our study shows that interannual variability of SEB at the SFP is relatively weaker because of artificial management. For better readability, we revised our manuscript.

L235-7 This discussion of QE fits better with the following evapotranspiration paragraph. Perhaps worth mentioning somewhere that QE and ET are equivalent – currently they are discussed almost as two separate variables

**Reply:** We revised the texts to incorporate the reviewer's comment.

L238-51 Perhaps a sentence or two could be added to strengthen the discussion by considering sub-monthly variation with respect to the timing of the rainfall in July 2017. It would also be helpful to add some comparisons with other urban and forest sites in the literature and potential reasons for differences (e.g. as for the Fc discussion).

**Reply:** Figure 2, 4 and 5 show monthly variations during the measurement period and we revised the texts for better readability. We also revised our manuscript by adding comparison with other studies.

L245-6 The severity of the drought conditions should be made clearer in Section 2.2.2

**Reply:** We revised the text in Section 2.2.2 as the reviewer suggested.

L253-61 How were these warming rates calculated?

**Reply:** We removed these sentences because it is challenging to get statistically significant climatological values from relatively short-term temperature data of < 30 years.

The authors could consider swapping the order of Section 3.3 and 3.2. To me, it would seem more natural the other way around.

**Reply:** We revised the text as the reviewer suggested.

L262-9 Is this temperature difference significant? Are the results robust (e.g. for the three sites that make up AVG independently)? How were the measurement and elevation heights accounted for? How was it ensured that the differences seen are not due to differences between instruments?

**Reply:** Please consider that temperature sensors are calibrated every two years and that we are focusing on temperature difference before and after the park construction (i.e., height is not important because measurement height did not change at all during our analysis). Please also consider that because temperature decreases with increasing height typically and the measurement height at the CBD is higher than that at the SD, temperature differences between AVG (currently modified to CBD) and SD will be larger if we account for the height difference. We revised the texts for better readability.

L277 'A possible reason for this' – this should be stated more strongly as the whole of the temperature part of the of the paper is based on the local characteristics being responsible for the

near-surface conditions

**Reply:** We revised the text as the reviewer suggested.

L287-91 The thermal admittance discussion is not clear, please rephrase
(Our findings indicate that the urban forest has a similar air temperature in the daytime as compared to the rural area (i.e., GP) where has **a lower thermal admittance** because of its location within the airport.) Perhaps consider renaming 'AVG' to something more meaningful to readers, such as 'CBD' for Central 'Business District'

**Reply:** We removed this sentence for better readability and changed 'AVG' to 'CBD'.

L309-10 It's not clear which results are shown here and which are from Lee et al. (2021). Try to make this clearer so that the reader knows what they can learn from this paper, and what additional information they need to look in the other paper for

**Reply:** We revised the text as the reviewer suggested.

L315 A couple of references showing this in natural ecosystems would be helpful here

**Reply:** We added several references here as the reviewer suggested.

L328-9 It is difficult to see this relation in Fig 9. Perhaps delete and rely on the reference to Lee et al. (2021)

**Reply:** We revised the text as the reviewer suggested.

Fig 3 first seems to be referenced in L335, after Figs 4-10. Given the importance of land cover around the tower in this section of the analysis, it would be helpful to add a subsection to 2.2 where the site characteristics are clearly described in preparation for these results. There is some information spread through Section 2.2.1 at the moment but this should be extended and more clearly described so the reader gets a good understanding of the site early on. The analysis in L346-355 may fit better at the start of Section 3.4 as it gives a general overview of the different processes in different parts of the source area.

**Reply:** We revised the text and added a figure as the reviewer suggested.

L342 Should Fig 10b really be referenced here?

**Reply**: We revised the text as the reviewer suggested.

L344 Emissions from the park facility are mentioned here for the first time. It would be good to include this in the site description and to discuss the effect on the results. Are these emissions from a single nearby building responsible for a large proportion of the measured Fc, and if so, what implications would this have for the reliability of the measurements and the annual total CO2 flux? Is it possible the Tair and QE measurements were also affected by these emissions?

**Reply**: As the reviewer suggested, we revised our manuscript by adding information on building emissions in the site description. Please consider that there is only a building within the tower footprint and this building emission occurs only in winter because of hot water and space heating. Our estimations on anthropogenic emission from vehicle and building show good correlation with inventory data such as visitor counts, traffic volume, and natural gas consumption in the park (Lee et al., 2021). Also, we discussed that our partitioning indicated that building emission was important contribution to net CO2 exchange in winter (Fig. 13). Please also consider that air temperature is considered in the flux partitioning method.

L402-407 This paragraph is not very clear. Please rephrase

**Reply**: We revised the text as the reviewer suggested.

L415-516 The Fc results are set in context using other studies. It would be good to do the same for the UHI and energy balance results.

**Reply:** We revised our manuscript for comparison of our results with other studies. We added a table to summarize surface fluxes at several urban sites of the similar vegetation cover. We expanded our comparison of UHIi to the study in Korea by Hong et al. (2019a) that used the Gimpo airport weather station as the reference. Please also consider that it is hard to make general comparison with other previous studies because they used different rural or suburban sites when they calculated UHI.

L463-5 Here the text suggests that the flux partitioning method was proposed in this paper which is not the case. There is no description of the method in this paper. Please amend.

**Reply**: We revised the text as the reviewer suggested.

Similarly in L514-6 it should be made clear that the partitioning method is from the Lee et al. (2021) paper and not developed in the current paper.
**Reply**: We revised the text as the reviewer suggested.

L471-3 Consider mentioning irrigation here
**Reply**: We revised the text as the reviewer suggested.

L483-4 It is not clear if the 'rule of thumb' is for this study or more generally. Probably replace 'rule of thumb' by 'For this study'
**Reply**: We revised the text as the reviewer suggested.

Table 1 It would help the reader to write the descriptions of the LCZs here instead of subscripts with the codes which most readers would have to look up elsewhere. 'Flux' is too general in the 'used variables' column. Please be specific (or this column could probably be deleted). What to the heights represent – measurement height above ground, elevation?
**Reply**: We revised the table as the reviewer suggested.

Table 2 As mentioned above details are needed about the gap filling procedure. Do these totals correspond to the source area (i.e. dependent on the wind direction distribution) or neighbourhood averaged values? Were the energy fluxes here gap-filled analogously to the CO2 fluxes or were totally different methods used? Similarly for Fig 13, do these bars correspond to the source area composition (i.e. are they affected by wind direction) or has the wind direction dependency been accounted for by the gap-filling procedure.

**Reply**: We revised the table as the reviewer suggested by adding more information on the gap filling and flux partitioning method.

Figure 3 How was the dominant wind direction decided? Is this the modal value? Change 'ratio' to 'proportion'. Currently Fig 3 is not used very much in the text – as suggested above developing this

part to give a clearer idea of source area characteristics and variability would be helpful for the reader. As an example, the proportion of road in the footprint could be added.

**Reply**: Our estimation is based on wind rose analysis and we replaced this figure with a wind rose

.

Fig 4 Consider adding a row with incoming shortwave measurements as you have the data.
**Reply**: We revised the table as the reviewer suggested.

Fig 8 It would be interesting to explore the scatter in this graph and indicate which points correspond to which month in which year. What does the dashed line represent? Is a simple linear regression of y against x appropriate when both x and y have appreciable uncertainties? Why are there no error bars shown in the x-direction?
**Reply**: We revised the table as the reviewer suggested. We added error bars in x axis and applied for a linear regression to consider uncertainties in x and y.

Fig 9 The caption is difficult to follow and should be rephrased. The gap due to power failure should be mentioned in the Methods section.

**Reply**: We revised the table as the reviewer suggested.

Fig 10 Why was this monthly separation chosen between panels (a), (b) and (c)? It might help to indicate the road sector on this plot.

**Reply**: We revised this figure by adding the road sector. Please consider that there are apparent changes in each component in carbon balance equation on monthly basis.

Fig 11 It would be useful to add why EB does not feature in the equation in L834

**Reply**: Please consider that EB occurs only in winter because of hot water and space heating supported by natural gas consumption as we mentioned in our manuscript. Please also consider that light response curves of NEE and GPP are comparable to other vegetative canopies. We added this information in this figure caption for better readability.

Fig 12 It is not clear what the reference means here. What do the dashed lines represent?

**Reply**: We discussed this issue in the main texts and added the information into the figure caption too.

We revised all the very minor and language issues below as the reviewer suggested. We appreciate the reviewer's efforts for our manuscript. Thank you four your kind work for our manuscript.

Very minor/language issues

Generally the standard of English is very good, though there are a few places where more natural phrasing could be used. I have made some suggestions here:

L9 'two years of surface fluxes'

**Reply**: corrected.

L18 'than for typical'

**Reply**: corrected.

L21 'if the goal is lower CO2 emissions…' or 'when aiming to reduce CO2 emissions…'

**Reply**: corrected.

L24 'Cities comprise' or 'Cities make up'

**Reply**: corrected.

L31 'urban forests'

**Reply**: corrected.

L35 'during urban redevelopment'

**Reply**: corrected.

L36-7 'and overcome their maintenance costs'

**Reply**: corrected.

L39 'have been addressed'

**Reply**: corrected.

L43 'for longer than'

**Reply**: corrected.

L44 'contribute to reducing'

**Reply**: corrected.

L53 'have reported on the surface'

**Reply**: corrected.

L55-6 'Forests can even produce a warming trend as a result of their low albedo'

**Reply**: corrected.

L73-4 'It is challenging to partition Fc into individual sources and sinks in urban areas because'

**Reply**: corrected.

L78 'carbon cycles'

**Reply**: corrected.

L81-2 Not clear, could delete the text 'where a hot… …warming trends'

**Reply**: corrected.

L83 'based on partitioning of FC data measured by eddy covariance'

**Reply**: corrected.

L128 'south and west sectors (120-330°)'

**Reply**: corrected.

L129 'and roads lie outside of the park'

**Reply**: corrected.

L137 'The roads'

**Reply**: corrected.

L166 'for comparative analysis because the sites are'

**Reply**: corrected.

L207 Delete 'after the processes'

**Reply**: corrected.

L208 'Here we partition the measured Fc into'

**Reply**: corrected.

L209-10 'This study applied a statistical'

**Reply**: We revised the sentence including this line based on the major comments.

L250 'is unknown' or 'has not been quantified' would be clearer than 'is inexplicit'

**Reply**: corrected.

L267 'when photosynthesis is highest. Our results'

**Reply**: corrected.

L271 'also produces'

**Reply**: corrected.

L309 Define EVI

**Reply**: defined.

L310-1 Move 'accordingly' to the start of this sentence

**Reply**: corrected.

L312-3 'absorbs more $CO_2$ than is emitted by local sources and $F_c$ is negative only during the summer daytime. Because of...'

**Reply**: corrected.

L320 'Greater reduction'

**Reply**: corrected.

L332 'The seasonal $F_c$ variation also depends on...'

**Reply**: corrected.

L335 Delete 'particularly'

**Reply**: corrected.

L348 'further indicate the'

**Reply**: corrected.

L359 'contrast to $F_c$ in high-rise high population residential areas... that does not respond to PAR'

**Reply**: corrected.

L318 'from a suburban area with about'

**Reply**: corrected (L381).

L425 Delete 'cumulative sum of the'

**Reply**: corrected.

L456-7 'method in order to examine the role'

**Reply**: corrected.

L462 Suggest deleting 'and was redeveloped from a racetrack and factory in the mid-2000s' as this is not really relevant for the conclusions

**Reply**: corrected.

L479 'a typical diurnal'

**Reply**: corrected.

L418 'and the time of the minimum is delayed'

**Reply**: corrected (L481).

L500 Delete 'Eventually'

**Reply**: corrected.

L508 'forests have important impacts on air...'

**Reply**: revised.

L810 Move 'in summer' after 'AVG and SD'

**Reply**: corrected.

---

## Author Comment (AC2)

**Review of ACP-2021-354**

Overall, I think the authors have written a nice study on the meteorology, energy balance, and CO2 fluxes of an urban forest in South Korea. I found the study informative and interesting to read. I have a few suggestions on the manuscript; most of these suggestions relate to clarifying the text and fixing grammar in a few places. Note that, although I am a CO2 flux modeler, I am not a technical expert on eddy flux measurements. Hence, I do not have many technical comments on the actual measurement or partitioning approaches used in this study.

**Reply:** Thank you for your constructive and critical review and we appreciate your effort. We tried to revise our manuscript based on the reviewer's comments.

- Line 17: I'm not sure that "stipulate" is the right word here. Perhaps "find" or "hypothesize"?

**Reply**: We revised the word as the reviewer suggested.

- Line 25 "our life trajectory": Whose life trajectory are you referring to here? Are you referring to the life trajectory of all humans? Maybe a different phrase would be better here (e.g., "human civilization heavily depends ….").

**Reply**: We revised the word as the reviewer suggested.

- Abstract and intro: The authors use the word "our" frequently in the text. Sometimes, I think the authors use this word to refer to themselves, and sometimes I think the authors use this word to refer to all of humanity. I would try to be clearer or more careful with the use of "our".

**Reply**: We carefully checked this word in these parts.

- Lines 33-38: This text feels generic and vague. I would cut these sentences from this paragraph.

**Reply**: We revised the word as the reviewer suggested.

- Line 82: I don't think the verb "affects" works in this sentence. I would delete it and replace "affects and shows" with "mirrors a steep global warming trend."

**Reply**: We revised the word as the reviewer suggested.

- Line 159: I would remove the word "additionally". It's not clear from this sentence what these observations are in addition to.

**Reply**: We revised the word as the reviewer suggested.

- Line 247 "similar net radiation": Did radiation show a similar decrease as precipitation, or is radiation in the second year similar to the first year? This distinction wasn't clear to me from the text.

**Reply**: We revised the word as the reviewer suggested for better readability.

- Line 253 "Figure 6 shows….": This sentence doesn't feel like a very effective topic sentence. Instead, I would start with a sentence that summarizes the main result of Fig. 6.

**Reply**: We revised the word as the reviewer suggested.

- Line 271: I would replace the word "produces" with "is".

**Reply**: We revised the word as the reviewer suggested.

- Line 306: I think you could also construct a more effective topic sentence here. I would start out with the main conclusion about temporal dynamics of net CO2 exchange and then refer to Fig. 9 to support that conclusion.

**Reply**: We revised the word as the reviewer suggested.

- Line 324 "As photosynthesis decreases, FC changed": The two verbs at the beginning of this sentence have different tenses ("decreases" versus "changed"). I would use consistent tense.

**Reply**: We corrected it.

- Line 332 "With such apparent seasonal FC variation, it is notable that its…": What does "its" refer to in this sentence?

**Reply**: We revised the word as the reviewer suggested.

- Line 333: What do you mean by "flux source area"? Are you referring to different land cover or land use types? Overall, I'm confused by this sentence. The sentence seems to say that variability in CO2 fluxes depends on spatial-temporal variability in CO2 fluxes.

**Reply**: In flux measurements, flux values change with measurement footprint. We revised the word as the reviewer suggested for better readability.

- Throughout the results and discussion, there are a lot of abbreviations. On one hand, I think those abbreviations are very useful to avoid repeating longer phrases. On the other hand, there were several instances when I had to look up numerous abbreviations to understand a topic sentence. Lines 340 - 341 are a good example. In these abbreviation-heavy sentences, I wonder if it's possible to state the conclusion in simpler terms and subsequently bring in technical

abbreviations to illustrate that point.

**Reply**: By our mistake, we omitted an appendix for the abbreviation. We added the abbreviation in the manuscript. We revised the manuscript as the reviewer suggested and please check our revision carries simpler conclusion.

- Line 390: I think this topic sentence could be strengthened (as per previous suggestions).

**Reply**: We revised the word as the reviewer suggested.

- Line 411 "and similar to that in the urban forest in East Asia ": What urban forest are your referring to here? Or are you referring to urban forests more broadly?

**Reply**: The urban forest studies are rare, and we cited natural forests in the similar climate zone. We revised the word as the reviewer suggested for better readability.

- Line 425: See previous suggestions about strengthening topic sentences.

**Reply**: We revised the word as the reviewer suggested.

- Lines 459-460: Awkward wording. This sentence also feels out of place in this paragraph because it describes results, whereas other sentences in the paragraph describe the measurement site and site context.

**Reply**: This sentence indicates climate conditions during the measurement period and accordingly, we may be able to retain this sentence but change its position.

- Line 508 "make significant traces": I'm not sure what this phrase means. How about "has a significant effect on".

**Reply**: We revised this sentence for better readability as the reviewer suggested.

---

## Author Response (AR2)

Suggestions for revision or reasons for rejection (will be published if the paper is accepted for final publication)

This is an interesting and useful study of the urban heat island, energy fluxes and CO2 exchange at a complex urban forest site in Seoul, Korea. Spatiotemporal trends in the observed fluxes are analysed and related to the characteristics of the site and climate conditions (particularly the effect of the East Asian monsoon and a drought period in one of the study years). Urban-rural differences in CO2 exchange and the role of the urban forest in reducing the urban heat island are discussed. The comparison with other urban sites nearby is interesting and discussion incorporating previous studies in the literature helps to set this work in context. The findings of this study are useful for urban planning applications and raise the important point that the potential impact of land management and weather conditions on urban green spaces need to be considered when designing cities.

The authors have addressed the reviewer comments well. The summary of key information from previous studies about the gap-filling methods, flux partitioning and calculation of displacement height and roughness length is much improved. I have a few suggestions for further improvement and recommend this article for publication following consideration of these minor points.

Response: We thank reviewers and editor for their constructive comments on our manuscript, which undoubtedly improved our manuscript. We did our best in revising our manuscript based on all the reviewer's comments and please check our responses below.

Minor revisions

To improve traceability, it would also be good to mention which of the methods in Kent et al. (2018) were used to inform the values for roughness length and displacement height given in L140-144.

Response: We revised our texts as the reviewer suggested by adding the method used in this study.

It is still not very clear how the authors arrive at the conclusion that the tower height requirement is satisfied for most wind directions (L158-160), nor how severely the requirement is not met for particular wind sectors. It would be good to add a sentence here stating which wind directions are potentially problematic (citing results from Kent et al. (2018) if necessary), and if possible, to give an indication to the readers of how this might impact the results.

Response: We added more information into the revised manuscript for the reviewer's comment by indicating the potential issue in a particular wind direction.

L110 Section 2.2.1 presents the climate conditions before the reader knows where this study takes place. Where are the climatological mean values in L112 measured? Similarly, the characteristics of the flux footprint are discussed before the location of the flux tower has been introduced. Suggest moving Section 2.2.1 after Section 2.2.2 (as in the previous version) or even to the end of Section 2.

Response: We revised our manuscript as the reviewer suggested.

L116-118: Swap the order of these two sentences to talk about general conditions first and then the effect on the flux footprint.

Response: We revised our manuscript as the reviewer suggested.

L205: The quality control procedure removes negative CO2 fluxes during night-time on the basis that there is no photosynthesis at night. Presumably the explanation for these negative fluxes is random error and the expected flux should be close to zero. If so, doesn't removing only negative values introduce a bias into the dataset, since the small positive CO2 fluxes are not removed and so the daily/seasonal/annual sum will be larger than expected? Perhaps the authors could add a comment on whether these negative values are substantial proportion of the dataset or only occur occasionally, and what the likely impact of this particular quality control test on the annual total fluxes is.

Response: This is an important point and we added information on the percentage of negative nocturnal $F_C$ values (n = 485, 1.4%) and their potential impact on the annual net carbon exchange (-0.0075 $kgCO_2$ $m^{-2}$ $year^{-1}$, 1.4% of annual $F_C$) into the revised the manuscript. We considered that such negative nighttime $F_C$ values is beyond the randomly generated error because their standard deviation is relatively larger than that in positive nighttime values. We also considered that generally random error $F_C$ of is relatively smaller in positive signs than negative signs based on the uncertainty analysis (i.e., asymmetry) (e.g., Hong et al., 2020, Int. J. Clim).

L467-470: You could add the reference here to make it clear that the gap-filling and partitioning are based on methods in a separate publication.

Response: We revised our manuscript as the reviewer suggested.

Fig 10: A minor issue, but it is still not clear to me why the panels group different numbers of months together in this way (Jun-Sep, Oct-Feb and Mar-May). Seasonal variation is considered in other figures and also discussed in the text, so perhaps four panels with Jun-

Aug, Sep-Nov, Dec-Feb and Mar-May would fit better, or three panels with four months each, or even two panels separated into the growing season (e.g. May-Oct) and non-growing season. The authors could consider whether a different grouping of months may fit better with the text.

Response: We revised Fig. 10 and its related texts by separating two seasons, (a) growing season (May-Oct.), (b) non-growing season (Nov.-Apr.) as the reviewer suggested.

[Figure]

The manuscript would benefit from further English language editing to improve readability. I have made some suggestions below but there are many more which should be corrected before publication.

L48: change to 'at scales from street trees to parks'

Response: We corrected the text.

L94: delete 'from soil and vegetation' as this contradicts the following sentence

Response: We corrected the text.

L96 change to 'park visitors'

Response: We corrected the text.

L128: I think this should be Fig 1 not Fig 2.

Response: We corrected the text.

L251: delete 'over'

Response: We corrected the text.

L283: change 'less' to 'lower'

Response: We corrected the text.

L335: change to 'downward shortwave radiation'

Response: We corrected the text.

L373: change to 'trends with PAR'

Response: We corrected the text.

L795: The Basel sites have much lower vegetation fraction (seems strange to describe them as 'similar').

Response: We corrected the text.

L828: delete 'surrounding'

Response: We corrected the text.

L829: change to 'around the flux tower'

Response: We corrected the text. Thank you.